# Does darkness increase the risk of certain types of crime? A registered report article

**Jim Uttley** [1]*, **Rosie Canwell** [2], **Jamie Smith** [2], **Sarah Falconer** [2], **Yichong Mao** [1], **Steve Fotios** [1]

**1** School of Architecture, University of Sheffield, Sheffield, United Kingdom, **2** South Yorkshire Police, Sheffield, United Kingdom

* j.uttley@sheffield.ac.uk

## Abstract

Evidence about the relationship between lighting and crime is mixed. Although a review of evidence found that improved road/ street lighting was associated with reductions in crime, these reductions occurred in daylight as well as after dark, suggesting any effect was not due only to changes in visual conditions. One limitation of previous studies is that crime data are reported in aggregate and thus previous analyses were required to make simplifications concerning types of crimes or locations. We addressed this issue by working with a UK police force to access records of individual crimes. We used these data to determine whether the risk of crime at a specific time of day is greater after dark than during daylight, using a case and control approach to analyse ten years of crime data. We compared counts of crimes in 'case' hours, that are in daylight and darkness at different times of the year, and 'control' hours, that are in daylight throughout the year. From these counts we calculated odds ratios as a measure of the effect of darkness on risk of crime. The results supported our three hypotheses: 1) The risk of overall crime occurring after dark was greater than during daylight (OR: 1.28, 95%CI: 1.23–1.34); 2) The risk of crime occurring after dark varied depending on crime category, with five out of fourteen crime categories having odds ratios greater than 1.0; and 3) The risk of crime occurring after dark varied depending on geographical area, with 25 out of 172 Middle Super Output Areas in South Yorkshire having odds ratios greater than 1.0. Our results suggest darkness increases the risk of Bicycle Theft, Burglary, Criminal damage, Robbery – personal, and Vehicle offences, and that some areas may be at more risk of crime occurring after dark than others. These findings suggest the crime types where outdoor lighting may help reduce the risk of crime after dark.

**Data availability statement:** The raw data used in the main and sensitivity analyses cannot be made available as these data are held by South Yorkshire Police, and the details included in the data, such as time and date, crime category, and location, could be used to identify victims. However, we provide sets of synthetic data that reflect the data structure and variables of the real data, as Supplementary files S2 and S3. We provide output data in the form of odds ratios and 95% confidence intervals for each crime category in Supplementary file S4 (main analysis) and S5 (sensitivity analysis). These data are also provided as Tables 7 and 8 in the main paper. We provide output data in the form of odds ratios and 95% confidence intervals for each MSOA in Supplementary file S6 (main analysis) and S7 (sensitivity analysis). We provide output data in the form of odds ratios and 95% confidence intervals for null tests 1 and 2 in Supplementary files S9 and S10.

**Funding:** The author(s) received no specific funding for this work.

**Competing interests:** The authors have declared that no competing interests exist.

# 1. Introduction

## 1.1. Darkness, lighting and fear of crime

Darkness is likely to influence perceptions of safety and fear of crime amongst pedestrians. Previous studies have found significant differences in subjective evaluations of safety and fear of crime under different conditions of ambient light – i.e., daylight versus darkness [e.g., 1–3]. These differences have been found in a range of different types of study design. For example, Gover et al [1] asked staff and students on a University campus about their fear of crime and perceived risk of crime during the day and at night. Fear of crime and perceived risk of crime were higher at night than during the day. Some studies show participants photographs or virtual reality images representing daylight and after dark conditions, and ask about perceptions of safety. In one study, Loewen, Steel & Suedfeld [2] showed participants a series of photographs of scenes that were in daylight or darkness:daylight scenes produced significantly higher ratings of safety than those scenes shown in darkness. Other studies have taken participants to real locations in both daylight and after dark and asked them to evaluate how safe they felt in that location. Boyce et al [3] took participants to 24 open parking lots and found that ratings of how safe it felt to walk at those parking lots were consistently lower during the after-dark visit than during the daylight visit. A similar method was used by Fotios, Monteiro & Uttley [4] who took participants to ten locations in an urban residential area. Ratings of safety during the after-dark visits were significantly lower than during the daylight visits. Research also shows a relationship between the level of light provided by street lighting after dark and feelings of safety. The study by Boyce et al [3] for example showed that as the median illuminance of a car park increased the difference between daylight and after dark ratings of safety reduced, suggesting participants felt safer as illuminance increased. The study by Fotios, Monteiro & Uttley [4] demonstrated a similar relationship between illuminance and feelings of safety.

In summary, a range of studies using different methods have demonstrated perceptions of safety tend to be lower (and fear of crime higher) after dark than during daylight, and when an area is less well lit after dark. One reason why it feels less safe after dark is that visual function is impaired at low light levels [5] meaning it becomes slower and more difficult to detect and identify visual features of the environment. This can limit the ability to make judgements of prospect and refuge, which are known to influence how safe we feel [6].

## 1.2. Darkness, lighting and actual crime

In contrast to perceived risk of crime and perceptions of safety, the relationship between light levels and actual crime and safety is less clear. Evidence about offender perceptions and their decision-making processes does not reveal a consistent influence of darkness or daylight – this influence is mediated by other factors such as victim type and type of crime. In interviews with sexual assault offenders, Balemba and Beauregard [7] found that offenders were more likely to commit a sexual assault after dark if their victim was an adult, but a younger victim was more

likely to be attacked during daylight. Palmer, Holmes and Hollin [8] found that domestic burglars had no clear preference for when they committed their offence – a third preferred the morning (presumably in daylight although this is not stated) and a third preferred after-dark. In contrast, Tabirizi and Madanipour [9] found that domestic burglars did prefer to commit burglary after dark as it reduced the chance of being seen by neighbours and passers-by, and made it easier to see whether occupants were at home or not. Street robbers might be expected to prefer to commit robbery after dark, as there are fewer people around to intervene and they will be less identifiable to their victim, but interviews with convicted street robbers suggested there was no general preference for committing robbery after dark – offenders tended to be active at all times of the day [10]. Findings from interviews with offenders do not therefore suggest a clear preference for committing crime after dark, and if there is any preference this is likely to be mediated by other factors such as the type of crime and victim characteristics.

Evidence from data about committed crime also fails to reveal a clear pattern in relation to light conditions. Steinbach et al [11] examined how Local Authority street lighting strategies for energy saving such as dimming and partial switch-off influenced crime counts. They found no clear relationship between any of the lighting strategies examined and the numbers of crimes recorded. The study has limitations however that raise a question over this conclusion (e.g., see [12]). For example, only 62 out of 174 Local Authorities in England and Wales provided data for the study: A self-selection bias may have occurred, with Local Authorities more likely to provide their data if they anticipated no detrimental effect on crime rates of their practise of lighting switch-off or dimming. The crime data used in the study did not include information about the time of occurrence meaning it was not possible to distinguish between crimes that occurred after dark or during daylight. In an attempt to address this, the authors of that work included only crime types that were believed more likely to occur after dark (e.g., burglary, theft from vehicle, robbery, sexual assault). Further analysis of the relationship between switching off street lights and crime rates found that theft from vehicle offences reduced when street lights were switched off after midnight, an opposite effect to the assumed benefit lighting has on the occurrence of crime [13]. Spatial displacement was found, with an increase in theft from vehicles in adjacent areas where street lighting remained unchanged, further evidence that the presence of lighting could increase rather than decrease certain crimes. Davies and Farrington [12] examined the effect of switching off street lighting at night (between 23:30 and 05:30) on crime. They compared one district that carried out this switch-off with another district that did not. Results were mixed – burglary and vehicle crime reduced in the switch-off district but not as much as in the control district; however, violent crime reduced significantly more in the switch-off district compared with the control district.

A systematic review of the effect of lighting interventions on crime rates found that improved street lighting (for example newly-installed or brighter lighting) was associated with a reduction in crime [14]. However, reductions in crime were found in daylight as well as after dark, suggesting any effect of the lighting was not due to improvements in visibility in an area but instead to an increase in 'community pride' – an area receiving local government attention in the form of street lighting improvements prompts increased community cohesiveness, informal social control and more people outdoors, with more eyes on the street being the cause of a reduction in crime. The review by Welsh and Farrington has been criticised on statistical grounds [15,16]. One example of such criticism is that areas may receive investment to improve lighting because they have high levels of crime. Regression to the mean implies that crime levels would have reduced anyway in these areas, regardless of any lighting intervention.

Welsh and Farrington recently updated their review of the effect of street lighting on crime rates [17]. This new review included the original thirteen studies from their earlier review along with eight new studies published since the earlier review that met the inclusion criteria. The new review concluded that there is still a beneficial effect of street lighting on crime based on the evidence, with crimes decreasing by 14% in treatment areas that receive improved street lighting compared with control areas that do not. However, this includes crimes committed during daylight as well as after dark. When only studies that included after-dark crime in their analyses were included in the meta-analysis, a non-significant 3% decrease in crimes in treatment areas was found.

One of the key studies included in the second Welsh et al review [17] was a randomised controlled trial of the effect of street lighting on crime [18], the first of this type of study on the topic. Mobile lighting towers were deployed in 40 randomly-selected housing developments and crime counts were compared with 40 control housing developments where the lighting towers were not deployed. Both groups of housing developments had high baseline levels of crime which helped overcome the issue of regression to the mean highlighted by Marchant [15]. Results suggested the lighting intervention led to a 35% reduction in serious crime after dark in the 6 months after the lighting deployment. It is not known whether this reduction in crime continued after 6 months though. A further aspect of the study to note is that the lighting intervention used in this study was not typical street lighting but mobile towers powered by a diesel generator. Each tower provided 600,000 lumens – between 17 and 120 times greater than the lumens provided by existing outdoor lighting in the housing developments. They therefore provided better visibility and were more obvious interventions than typical street lighting.

The relationship between street lighting and crime rates remains unclear – previous studies show mixed or weak results, and have a range of limitations. To help clarify the relationship between lighting and crime rates it is first useful to understand whether extreme differences in ambient light levels – daylight versus darkness – produce differences in crime. If such an extreme difference in ambient light levels shows no influence on crime rates, it seems unlikely that smaller changes in light levels brought about through lighting strategies or interventions can have any significant impact on crime rates, assuming visibility is the key pathway for any effect. A comparison of ambient light conditions can be done when there is no lighting intervention involved. Therefore, if any effect is found, it could be attributed to changes in visibility rather than changes in community pride caused by investment in an area. Distinguishing between the crime reducing effects of visibility and community pride is useful because there may be better ways than improving street lighting for enhancing community pride, perhaps methods which do not consume energy in operation or impact upon the nocturnal environment. The size of any effect found can also provide a useful measure of the effectiveness of any outdoor lighting strategy to reduce crime after dark. Effective lighting for crime reduction would reduce the risk of crime occurring after dark compared with during daylight in an area.

To measure the effect of darkness and lighting on crime it is necessary to account for other contributory factors that may act as confounds. It is insufficient to simply compare crime counts during periods of daylight with periods of darkness as a range of other confounding factors will also influence crime occurrence in addition to any effect of the ambient light condition. These include environmental factors such as shop density and house prices [19], socio-economic factors such as unemployment rates and income levels [20], and temporal factors such as time of day (e.g., [21]) and seasonal changes in weather conditions (e.g., [22,23]) which may affect the routine activities of people and motivation of offenders. For example, Herrmann [24] describes daily temporal patterns for murders, shootings, assaults and robberies in the Bronx, New York. There are large differences in the numbers of crimes depending on the time of day, with late evening generally showing peak crime counts. However, the peak hour for robberies is 15:00. Schools tend to finish around this hour, and further analysis of crime records showed that this hourly peak in robbery was largely caused by robberies involving school-aged offenders near schools and subway stations. These results show how the opportunity and motivation for crime vary depending on the time of day, as would be predicted by routine activity theory [25]. This highlights how the time of day influences the likelihood of crime occurring, including the type of crime that is committed.

One approach to overcoming confounding factors such as time of day and weather conditions is to use daylight saving time clock changes to create a natural experiment where time of day and other seasonal influences can be controlled whilst varying the ambient light level. At a Spring clock change, clocks are moved forward one hour, leading to a sudden, additional hour of daylight in the evening (whilst an hour of daylight in the morning is lost). At an Autumn clock change this is reversed, with an evening hour of daylight being lost as clocks are shifted back one hour. Comparing crime rates in the weeks immediately before and after a clock change, particularly around the hour of sunrise or sunset, can say something meaningful about the effect of ambient light conditions. Doleac and Sanders [26] carried out such an analysis of crime

records in the United States. The shift from darkness to daylight in the sunset hour following the Spring clock change was associated with a 27% decrease in robberies and a 38% decrease in rapes. This study primarily analysed crime data from less populous areas however, and the timing of crimes may vary between rural and urban areas.

The one-hour shift in evening daylight caused by the daylight saving time clock changes was also exploited by Fotios, Robbins and Farrall [27] to investigate the influence of ambient light on crime counts. They used a case/ control method to isolate the effect of darkness on crime counts. A 'case' window of time was selected that was in darkness before a clock change but was in daylight after the clock change (for a Spring clock change; this was reversed for an Autumn clock change). Counts of crimes that occurred in these two windows were compared in the week before and after the clock change. The ratio of counts between the after-dark and daylight case windows were then compared to a ratio of counts over a similar period but in control windows – periods of time that had the same duration as the case window but that occurred either two hours earlier or later so they had the same light condition (daylight or darkness) both before and after the clock change. The comparison of these two ratios was calculated as an odds ratio, where an odds ratio significantly greater than one would indicate crime was more likely to occur after dark than during daylight. Fotios, Robbins and Farrall [27] applied this method to crime data from three cities in the United States for the period 2010–2019. Results suggested a statistically significant effect of darkness on robbery but not on other types of crime. In a subsequent article, Fotios, Robbins and Farrall [28] extended this work by examining the influence of ambient light on crime in eight new cities in the United States, in addition to the three from the first study. This further supported a consistent effect of light on robbery, with this type of crime increasing when it was dark, but no other significant effects were found for other crime types. An effect of darkness on robbery has also been shown in other work that compared counts of robberies in 6-hour periods of the day that varied in terms of the proportional amount of darkness they contained [29].

The case/ control method used by Fotios, Robbins and Farrall [27,28] helps to isolate the effect of darkness by controlling for other confounding factors that are likely to influence crime rates, such as the time of day and weather conditions. However, one limitation of using the clock change to produce a sudden shift in ambient light conditions is the period of time in which crimes are included in the analysis is relatively small (in the analysis by Fotios, Robbins and Farrall, crimes were included if they occurred the week either side of each clock change, over a ten-year period, resulting in 40 weeks of crime counts included in their analysis). This limited time period can result in relatively low crime counts being used in the odds ratio calculation, producing large confidence intervals, particularly when analysed by crime category. An alternative approach that still uses the case/ control method to isolate the effect of darkness is utilising a whole year approach – selecting a case hour that is in daylight for part of the year and in darkness for another part of the year. This is possible for locations that are sufficiently North or South of the equator to show a large seasonal variation in daylight hours. In Sheffield, United Kingdom, for example, the hour of 19:00–19:59 will be in darkness between January-March and October-December, but in twilight or daylight for the rest of the year. A control hour can also be selected that remains in daylight (or darkness) throughout the whole year. This method allows recorded crimes from across the whole year to be included in the odds ratio calculation, rather than just in a short window around each clock change, reducing the associated confidence intervals. This whole year approach has been applied in work assessing the effect of darkness on walking and cycling rates [30,31], but not to assess the effect of darkness on crime rates.

## 1.3. Variations in effect of darkness and lighting on crime

Any influence of darkness on the risk of a crime occurring is unlikely to be equal for all types of crime. The committal of certain crimes may be more dependent on visibility levels than others, for example. Consider robbery – this is an interpersonal crime that requires the offender to get close to the victim. One factor that may deter a potential offender is the risk of being identified [32] – this risk reduces when it is dark and visibility is lower. The results of Fotios, Robbins and Farrall [27,28] suggested darkness increased the number of robbery crimes but not other types of crimes. By contrast, burglary does not always rely on ambient light levels – some residential dwellings are an excellent target in the daylight rather than

after dark because of their unoccupied status [33]. Disaggregating crime counts by the type of crime reduces the sample included in the odds ratio calculations, and the relatively low counts in some crime categories may account for why crimes other than robbery were not significantly affected by ambient light. Fotios, Robbins and Farrall [27] extrapolated their data from the three cities included in their analysis to calculate estimated odds ratios for the whole of the United States for different crime types. The extrapolated larger samples of crime counts suggested significant effects for other types of crime, such as destruction of property, theft and drunkenness. These conclusions were, however, based on extrapolated rather than real data.

The influence of darkness on crime risk may vary spatially. Certain streets or neighbourhoods may be more susceptible to after dark crime than others. This could be due to environmental factors – the physical form of the area, or a lack of adequate outdoor lighting, may reduce visibility after dark making some crimes more attractive to perpetrators. The risk of after-dark crime may also vary between areas due to the propensity of certain types of crime in those areas. If specific crimes are more likely to occur after dark than during daylight, those areas that show high levels of those crimes will also show a higher risk of crime after dark. Assessing the risk of crime occurring after dark for specific localised areas also has the added benefit that the area will act as a statistical control for itself, as other contributory factors that could act as confounds, such as aspects of the physical environment and the demographics of the area will remain constant.

In summary, there is much evidence that shows a relationship between light levels and perceptions of safety or reassurance. Evidence about the relationship between lighting and actual crime is not as clear. Whilst systematic reviews [14,17] suggest 'improved lighting' (which can include an increase in the presence of lighting, such as illuminating a previously unlit route, or increases in brightness of lighting) is linked to reductions in crime there are statistical limitations to this evidence [15,16]. The mechanism behind any effect of lighting on crime is also unclear, as much of the evidence that shows any effect does so for crimes during both daylight and after dark. This suggests improvements to visual conditions cannot be the only cause of an effect of lighting and other factors are at work, such as increases in community pride due to visible investment in an area. A first step in bringing clarity to the question of whether lighting influences crime levels is to assess the impact of darkness on crime, relative to daylight. If no effect is shown this rules out improvement of visual conditions as a causal mechanism for any effect of lighting on crime, and suggests there is no need to identify optimal lighting conditions for the reduction of crime after dark. If an effect is shown however, the size of this effect could be used as a measure to help optimise lighting characteristics such as illuminance and uniformity. In the context of crime reduction, good lighting could be considered that which offsets any increase in criminal activity after dark.

### 1.4. Current focus

Previous studies have shown that darkness can increase the risk of crime [26–28]. These works had limitations though such as a focus on less populous areas [26] and relatively limited sample sizes due to only using single weeks on either side of a clock change [27,28]. There is also uncertainty about the timing of crimes included in the analysis of these previous works. Many crimes are aoristic, meaning an exact time of committal is not known, only a window of time within which it could have occurred. Knowing the timing of a crime relatively precisely is important when comparing crimes at the same period of the day under different ambient light conditions. The previous work by Doleac & Sanders, and Fotios, Robbins & Farrall, does not address this point and it is likely they included crimes that occurred outside the period of time they were targeting. Previous work has also not examined differences in the effects of light at a sub-district level, although Fotios, Robbins & Farrall do compare between cities.

In the current study we assessed the impact of darkness on crime levels using crime data for the South Yorkshire region of the United Kingdom, and we addressed some of the limitations from previous work highlighted above. We used the whole-year case/ control method that controls for potentially confounding factors that may influence crime occurrence but are not related to light levels, such as the time of day. This method increases the counts of crimes included in the analysis compared with the clock change method used by Doleac & Sanders [26] and Fotios, Robbins & Farrall [27,28]. This

reduced uncertainty in odds ratios when the analysis was disaggregated by crime type. It also reduced uncertainty when disaggregating by sub-district areas. We also applied a screening process to ensure aoristic crimes were only included in the analysis if we could be relatively certain they occurred in one of our case or control periods.

We tested the following hypotheses:

1) The overall risk of crime in South Yorkshire occurring after dark is greater than during daylight, after time of day and seasonal factors have been accounted for

2) The risk of a crime occurring after dark relative to during daylight will vary depending on the type of crime

3) The risk of a crime occurring after dark relative to during daylight is not uniform across Middle Super Output Areas (defined below) within South Yorkshire

We tested these hypotheses by calculating odds ratios for all crime aggregated across the entire area of analysis (1), odds ratios for individual crime categories (2), and odds ratios for sub-areas within the entire area of analysis (3).

The next section outlines the method and analysis used for testing the three hypotheses. As this work involved the analysis of secondary data, the template for preregistration of secondary data analysis provided by van den Akker et al [34] was completed and is included as Supplementary S1 File in our Registered Report Protocol [35].

## 2. Method

### 2.1. Definition of darkness and daylight

We defined darkness as being when the sun's altitude is at or below −6°. We chose this definition because this represents the transition between civil twilight (when the sun's altitude is between −6° and 0°) and nautical twilight (when the sun's altitude is between −6° and −12° - see [36]). Based on data from solar monitoring sites in the UK, as reported in Raynham et al. [37], the average illuminance when the sun's altitude is at −6° is 2.33 lx.

We defined daylight as being when the sun's altitude is at or above 0° [36]. This altitude represents the time of sunrise or sunset. Based on the solar illuminance data reported in Raynham et al [37], the average illuminance at this altitude is 509 lx.

The period when the sun's altitude is between −6° and 0° was defined as civil twilight and represents a transition between ambient daylight and ambient darkness.

### 2.2. Assessing the impact of darkness

To assess the impact of darkness on crime rates we can compare counts of crimes during periods of darkness against counts of crimes during periods of daylight. Darkness occurs at different times of the day to daylight however, and time of day acts as a significant confounding factor with this approach. We can therefore compare counts of crimes that occur within the same hour but across the whole year, choosing this hour so that for part of the year it is in darkness and part of the year it is in daylight. For example, in Sheffield (UK), the hour between 18:30 and 19:29 is entirely in darkness between 1st January and 6th March, and between 31st October and 31st December. Between the 31st March and 10th September the hour is entirely in daylight. For the remaining periods of the year at least part of the hour is in twilight. We can compare counts of crimes during this hour when it is in darkness with counts when it is in daylight.

Although the above approach removes time of day as a confounding factor, by using the same time of day for both periods of darkness and of daylight, it does not account for changes in weather conditions or other seasonal factors that may influence crime rates. For example, the period when the hour of 18:30–19:29 is in daylight also coincides with better weather conditions, compared with when the hour is in darkness. Other seasonal factors that could contribute to crime rates also vary between the periods of daylight and darkness. Holiday and vacation periods have previously been associated with changes in crime rates [38] and there is an extended school vacation during the daylight period. Therefore

to account for seasonal factors that may influence crime rates, counts of crimes can also be recorded for a 'control' hour that remains in the same ambient light condition throughout the year. For example, for Sheffield, the hour of 14:00–14:59 remains in daylight between 1st January and 31st December. Changes between the crime counts occurring during the case hour when it is in daylight and darkness can be compared against changes in the crime counts occurring during the control hour during the same two periods of time. This comparison can be done by calculating an odds ratio, as shown in Equation 1 [39], using four separate counts of crimes determined by whether they occurred in the case or control hour, and at what time in the year, as shown in Table 1. The odds ratio provides a measure of the effect of darkness on crime rates that accounts for both the time of day and other seasonal factors such as weather conditions and vacation periods. An odds ratio significantly greater than one indicates the risk of a crime occurring after dark is greater than during daylight. A confidence interval (CI) for the odds ratio can be calculated using Equation 2.

$$OddsRatio = \frac{CaseDark}{CaseDay} \div \frac{ControlDark}{ControlDay} \tag{1}$$

$$95\% \; CI = exp\left(ln(OddsRatio) \pm 1.96\sqrt{\frac{1}{CaseDark} + \frac{1}{CaseDay} + \frac{1}{ControlDark} + \frac{1}{ControlDay}}\right) \tag{2}$$

Where:

CaseDark = Count of crimes in case hour when it is in darkness

CaseDay = Count of crimes in case hour when it is in daylight

ControlDark = Count of crimes in control hour when case hour is in darkness

ControlDay = Count of crimes in control hour when case hour is in daylight

For the current analysis three 60-minute periods were selected as case hours and these were paired with three control hours, as shown in Table 2.

## 2.3. Crime data

This analysis used crimes recorded by South Yorkshire Police, whose jurisdiction covers the Local Authority areas of Barnsley, Sheffield, Rotherham and Doncaster in the United Kingdom.

The police record a crime through a variety of channels, most frequently through attending a report of an incident. A crime is submitted for recording either by the operator dealing with the incident or the officer attending the scene. The

**Table 1. Contingency table showing how the four counts used in the odds ratio calculation are determined.**

| | Crimes that occurred on dates when the case hour is in darkness | Crimes that occurred on dates when the case hour is in daylight |
|---|---|---|
| **Crimes that occurred in the case hour** | CaseDark | CaseDay |
| **Crimes that occurred in the control hour** | ControlDark | ControlDay |

**Table 2. Pairs of case and control hours used for main analysis.**

| Case hour | Paired control hour |
|---|---|
| 17:30-18:29 | 13:00-13:59 |
| 18:30-19:29 | 14:00-14:59 |
| 19:30-20:29 | 15:00-15:59 |

crime will be recorded by the Force Crime Bureau (FCB), a sub division of the control room, and then allocated to an officer to investigate further. The crime is recorded on the force crime system, CONNECT (formerly CMS), according to the crime recording processes outlined in the 'Home Office counting rules for recorded crime' (HOCR). All police forces in the UK adhere to the HOCR and the National Crime Recording Standards (NCRS). This is to ensure crimes are recorded and counted in a standardised manner to allow for, amongst other reasons, a comparative overview of crime rates at a national and subnational level. In recording crime, there is oversight of the accuracy of this process conducted within South Yorkshire Police via external and internal audit provided in part by inspections conducted by His Majesty's Inspectorate of Constabulary (HMIC). For a definition of crime categories, see the Home Office's crime recording rules for front line officers and staff [40].

The data used in this analysis was extracted from the CONNECT and CMS systems used by South Yorkshire Police to record crime incidents. Crimes recorded as taking place between 1st January 2010 and 31st December 2019 were included in the analysis. This time period covers the operation of two crime recording systems in South Yorkshire Police, CMS (Jan 2010 – Nov 2017) and CONNECT (Dec 2017 – Dec 2019). In both cases data are extractable via the SQL based system of Oracle BI. The data contains information about the crime category recorded. This categorisation is based on a hierarchy of offences laid down in the HOCR and is detailed by HMIC in their crime tree diagram (see Fig 1).

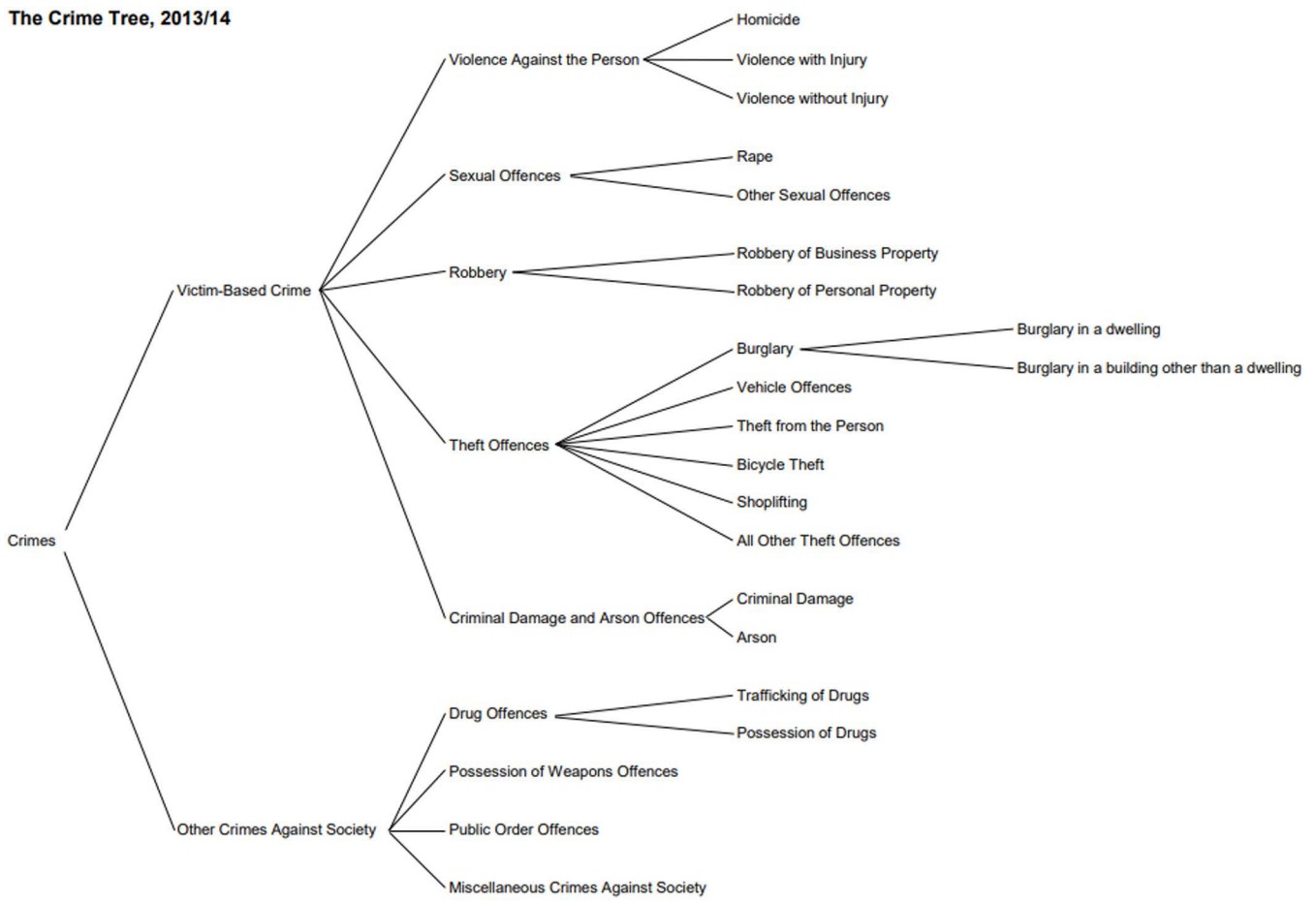

**Fig 1. The HMIC crime tree hierarchy with level 1, 2 and 3 categorisations.**

The study analyses crimes at the level 3 categorisation in the crime tree hierarchy (for the assessment of hypothesis 2), which strikes a balance of providing more detail to the nature of the crime versus the statistical power afforded/lost by aggregation/disaggregation.

Crimes that are classed as 'Other crimes against society' (also known as 'crimes against the state') were excluded from the data. These crimes were excluded as they can often be the result of police generated activity, i.e., the result of recording crime that wouldn't have previously been reported because of proactive patrols/targeted interventions. An example of this is drug possession offences. These are often the result of a search or a targeted patrol and as such the more that activity occurs, the more the police record, meaning the indicator is a measure of police activity and not a base level of 'true' criminality.

The crime data also includes the Middle Super Output Area (MSOA) where each crime took place. MSOAs are a layer of geographical areas in the UK designed to support the reporting of small-area statistics. MSOAs have a minimum population of 5,000, with a mean population of 7,200. As the definition of an MSOA boundary is based around population level they can vary greatly in area – being relatively small in densely populated areas but large in sparsely populated areas. Our analysis provided overall crime odds ratios for each MSOA in South Yorkshire. Odds ratios by crime type for each MSOA were not calculated because this level of disaggregation would lead to very small crime counts included in the odds ratio calculation, and potentially also produce ethical issues in terms of anonymity, with small count data potentially allowing the identification of victims.

A fictional example of unfiltered and unprocessed data used in this analysis is shown in Table 3. Note that a real example of the data cannot be shown due to ethical issues caused by the potential breach of victim anonymity and data confidentiality. The data includes details of crime locations and times, which could potentially identify individual victims.

## 2.4. Data analysis

The analysis calculated odds ratios to make inferences about the influence of darkness on the risk of crime. The data was first filtered to only include those crimes that occur in pairs of case and control hours (see Table 2). The data included information about the time and date the crime was committed. A 'Committed From Date' and 'Start time' are recorded. Our analysis script combined these to create a 'committed from' time and date variable. A 'Committed To Date' and 'End time' are also recorded, and these were combined to create a 'committed to' time and date variable. For crimes when the exact time of committal are known, these two time and date variables are identical. Some crime records do not include a value for 'End time', although the 'Committed To Date' is identical to the 'Committed From Date'. In these instances it was assumed that the end time is identical to the start time. Many crimes are aoristic. For such crimes, the record of the offence provides a window of time the crime was potentially committed in. Crimes where the time of committal is known but have taken place over a period of time, rather than at a specific instance, may also have a window of time when the crime was committed. It is not possible to distinguish between these and aoristic crimes however, so we therefore treat both types of crime in the same way. Only crimes that could have been committed in a case or control hour should be

**Table 3. Example of unfiltered, unprocessed fictional data.**

| Incident Number | HMIC Crime Tree Level 3 | MSOA | Committed From Date | Committed To Date | Start time | End time |
|---|---|---|---|---|---|---|
| 3 | ALL OTHER THEFT OFFENCES | Hackenthorpe | 22/09/2018 | 24/09/2018 | 17:12 | 03:41 |
| 4 | THEFT FROM THE PERSON | Handsworth South | 20/01/2018 | 22/02/2018 | 02:05 | 02:05 |
| 5 | BURGLARY – DWELLING | Lower Stannington | 23/03/2018 | 23/03/2018 | 18:12 | 21:53 |
| 6 | BICYCLE THEFT | Crabtree & Fir Vale | 31/10/2018 | 31/10/2018 | 16:56 | 19:00 |
| 7 | ROBBERY – BUSINESS | Southey Green West | 11/05/2018 | 11/05/2018 | 19:59 | 21:23 |
| 8 | ARSON | Sharrow | 14/02/2018 | 14/02/202018 | 18:22 | Not Recorded |

included in the analysis. Therefore two inclusion criteria were applied to aoristic crimes and crimes that are committed over a period of time, where an exact committal time is unknown, to determine their inclusion in the final dataset:

1) The midpoint between the 'committed from' and 'committed to' times should fall within one of the case or control hours (the midpoint within a crime time 'window' has been shown to be a better estimate of committal time than the start or end of that window [21])

2) The difference between the 'committed from' and 'committed to' times should be less than one hour

Crime records that met these two inclusion criteria were included in the final dataset for analysis. Crimes where the exact committal time is known (i.e., the 'committed from' and 'committed to' times are the same, or a 'committed from' time is given but not a 'committed to' time) were also included in the final dataset if this time fell within one of the case or control hours.

Counts of those crimes that fell within a case or control hour, based on the above criteria, were aggregated into four groups, based on when they occurred:

CaseDark: Crimes that occurred during one of the case hours when that case hour was in darkness.

CaseDay: Crimes that occurred during one of the case hours when that case hour was in daylight.

ControlDark: Crimes that occurred during one of the control hours when that hour's paired case hour was in darkness. For example, crimes that occurred during 13:00–13:59 would be included in this group if they occurred on a date when the paired case hour, 17:30–18:29, was in darkness.

ControlDay: Crimes that occurred during one of the control hours when that hour's paired case hour was in daylight.

Whether a case hour is in daylight or darkness was defined by day of the year, as set out in Table 4. On days of the year that fall outside the ranges for darkness and daylight shown in Table 4 the case hour will partially be in twilight. Any crimes that occured in case or control hours on such dates were excluded from the analysis.

Odds ratios were calculated using counts for CaseDark, CaseDay, ControlDark and ControlDay. However, the odds ratio calculation requires all four of these counts to be non-zero. If one of the counts was zero, 0.5 was added to all four counts used in the odds ratio, following the Haldane-Anscombe correction that is commonly used in such cases for odds ratios (e.g., see [41]). If more than one of the four counts was zero, the odds ratio was not calculated.

To assess hypothesis 1, that the overall risk of crime occurring after dark is greater than during daylight, the counts of all crimes were aggregated for each of the four periods and an overall odds ratio calculated from this, using Equation 1. The hypothesis is supported if this odds ratio is significantly greater than 1.0, based on its associated p-value, calculated using Fisher's exact test with the 2x2 contingency table of crime counts (see Table 1), with a criterion for significance set at 0.05. As we calculated a number of further p-values, associated with our testing of hypothesis 2, we used the Holm-Bonferroni correction to correct this p-value for multiple tests. The use of this correction was not included in our original analysis plan, described in the Registered Report Protocol.

To assess hypothesis 2, that the risk of a crime occurring after dark relative to during daylight will vary depending on the type of crime, the counts of crimes for each crime category were aggregated for the four periods and an odds ratio (using Equation 1) and associated 95% CI (using Equation 2) calculated for each crime category. The hypothesis is

**Table 4. Days of the year that define whether case hour is in daylight or darkness. Day 1 represents 1st January, day 365 (or 366 if it is a leap year) represents 31st December.**

| Case hour | Days of year case hour is in darkness | Days of year case hour is in daylight |
|---|---|---|
| 17:30-18:29 | 1-33, 304–365 (or 366 if leap year) | 87-278 |
| 18:30-19:29 | 1-65, 296–365 (or 366 if leap year) | 90-253 |
| 19:30-20:29 | 1-83, 270–365 (or 366 if leap year) | 120-227 |

supported if any of the CIs do not overlap. Non-overlapping CIs for two crime categories would indicate the risk of those crimes occurring after dark relative to daylight is not equal (e.g., see [42]). In addition, we calculated p-values associated with the odds ratio to assess whether they significantly deviated from one, using Fisher's exact test and corrected for multiple tests using the Holm-Bonferroni correction. Note that the calculation of these p-values was an additional step of analysis not included in our original Registered Report Protocol.

To assess hypothesis 3, that the risk of a crime occurring after dark relative to during daylight is not uniform across MSOAs in South Yorkshire, the counts of all crimes were aggregated for each of the four periods and a separate odds ratio and associated 95% CI calculated for each MSOA. The hypothesis is supported if any of the MSOA CIs do not overlap, as non-overlapping CIs for any two MSOAs would indicate the risk of crime occurring after dark relative to daylight is not equal between those MSOAs. In addition, we assessed whether any MSOAs had a higher (or lower) risk of crime being committed after dark than during daylight, by confirming whether their 95% CI did not overlap with one. Note that this step of analysis was an addition to what was originally described in the Registered Report Protocol for this study.

The analytical script and two synthetic (fictional, artificially generated) datasets (one representing the CMS dataset and one representing the Connect dataset) to be used with this script are provided as Supplementary files [S2]–[S4 Files]. These synthetic datasets have been generated randomly but reflect the structure, data types and variable headings that are present in the real data; note that an example of real data cannot be provided due to ethical issues caused by the potential breach of victim anonymity and data confidentiality, as time and location details are included which may identify a victim. The script produces an output showing the sum of counts of crimes in different case and control periods, either for different crime categories or for different MSOAs, depending on how the 'location' variable is defined in the script. The script also uses the output for the analysis by crime categories to calculate the corrected p-values associated with the odds ratios for each crime category and all crime categories combined.

## 2.5. Sensitivity analysis

We selected three pairs of case and control hours to use in this analysis. It is possible the results are sensitive to these choices, and selecting different pairs of case and control hours could yield different conclusions in relation to the three tested hypotheses. To assess this sensitivity to choice of case and control hours we also carried out a sensitivity analysis by using two alternative case hours from the morning, 05:00–05:59 and 06:00–06:59, paired with the control hours of 11:00–11:59 and 12:00–12:59 respectively. The days of the year that the two alternative case hours will be in daylight or darkness are shown in Table 5. This sensitivity analysis helps show whether the odds ratios reported in the main analysis are sensitive to the choice of case and control hours. This also helps show whether the time of day influences how darkness affects crime. Different types of crime tend to occur at different times of day [24] due to variation in structural opportunities to commit crime at different hours of the day. As well as being a validation of the main analysis, the sensitivity analysis helps show whether crimes committed at different hours are affected by darkness in different ways.

## 2.6. Statistical power

The epi.ssc function from the *R* package *epiR* was used to estimate the total count of crimes in the case and control periods required to detect odds ratios of sizes ranging between 1.2 and 2.5, with a minimum power of 80%, a confidence level

Table 5. Days of the year that define whether alternative case hours for sensitivity analysis are in daylight or darkness.

| Case hour | Days of year case hour is in darkness | Days of year case hour is in daylight |
|---|---|---|
| 05:00-05:59 | 1-67, 90-91, 257-365/366 | 141-196 |
| 06:00-06:59 | 1-39, 291-297, 325-365/366 | 84, 110-234 |

of 95%, and a one-sided test. These are shown in Table 6. In calculating these required counts the following assumptions were made, based on the data reported in Fotios et al [27]:

1) The number of crimes in the control hour when the case hour is in darkness is equal to the number of crimes in the control hour when the case hour is in daylight (i.e., bottom two cells in Table 1 are equal)

2) The total count of crimes in the control hour was 2.3 times that of the total count of crimes in the case hour

We did not know how much data was available to use in our analysis and what the frequency of counts in case and control hours would be, prior to undertaking this work, as the dataset was deliberately withheld (see section 2.7). However, we assessed whether sufficient counts of crimes would be recorded in order to make our analytical approach valid by making estimates based on known crime counts in South Yorkshire. Between 2009/10 and 2018/19 there were 990,446 crimes recorded in South Yorkshire [43], excluding those categorised as 'Other crimes against society'. Therefore 41,269 crimes were committed, on average in each hourly period across the entire 10-year period. A large proportion of these crimes will have a potential time range of when they could have been committed of more than an hour and would therefore not be included in our analysis. Data from Ratcliffe [44] suggests 51% of crimes have a time window of less than 4 hours. We therefore assumed that 20% of crimes will have a time window of one hour or less, and will be included in our analysis, although this assumption is not based on any specific rationale. This leaves a count of 8,254 crimes in each hour, on average, over 10 years. We included counts from six different hourly periods in our main analysis (three case hours and three control hours), giving a total count of 49,524. Based on the ratio of 2.3 crimes in a control hour for every crime in the case hour as found by Fotios et al [27], this suggests there could be 15,008 crimes in the case hours and 34,516 crimes in the control hours. Such counts allow an odds ratio of 1.05 or greater to be detected.

### 2.7.  Prior knowledge of data

Prior knowledge of the data used in this analysis has deliberately been limited. This was to avoid the opportunity to p-hack the data and obtain statistically significant but spurious results [45], and to limit the formulation of hypotheses that are already known to be in line with the data (Hypothesising After the Results are Known - HARKing - see [46]). Access to the data was provided by three of the authors (Smith, Falconer and Canwell) who are employees of South Yorkshire Police and had access to the crime records database from which data was extracted. However, the hypotheses and analytical approach were developed by two of the authors (Uttley and Fotios) who had no direct access to or prior experience of the data used. An analytical script was written in *R* by JU and SAF, and this was passed to Smith, Falconer and Canwell to apply to the crime data. We published a preregistration of our planned analyses prior to accessing any data – this was published as a Registered Report Protocol [35].

**Table 6.  Required total counts in case and control hours to detect different odds ratios with 80% power.**

| Odds ratio | Count required in case hour | Count required in control hour |
| --- | --- | --- |
| 1.2 | 1070 | 2461 |
| 1.5 | 219 | 504 |
| 1.8 | 105 | 242 |
| 2.1 | 67 | 155 |
| 2.4 | 49 | 113 |

## 3. Results

### 3.1. Main analysis

The filtering and inclusion criteria described in section 2.4 led to a total of 43,302 crimes being extracted from the crime databases, for the case and control hours used in the main analysis. This was relatively close to our estimate of 49,524 crimes, used for our power analysis (see section 2.6). However, we then had to exclude crimes that occurred on dates when the case hour was partially in twilight. This left 34,618 crimes to be aggregated for the main analysis. Twenty seven percent of crimes were in the CaseDark period, 24% in the CaseDay period, 23% in the ControlDark period and 27% in the ControlDay period.

Fourteen crime categories were included in the data, using level 3 categorisation of crimes from the HMIC crime tree diagram (Fig 1) and following exclusion of crimes classed as 'Other crimes against society' (see section 2.3). The counts of crimes in the CaseDark, CaseDay, ControlDark and ControlDay periods for these fourteen categories are shown in Table 7. The associated odds ratio, its 95% CI and p-value indicating whether the odds ratio significantly differs from 1.0 are also shown in Table 7. The p-values have been corrected using the Holm-Bonferroni method to account for multiple tests.

The overall combined odds ratio for all 14 crime categories included in our analysis is 1.30 (95% CI: 1.24–1.35, p<0.001) suggesting that darkness significantly increases the risk of crime occurring. Five individual crime categories also have odds ratios significantly greater than 1.0: Bicycle theft, Burglary, Criminal damage, Robbery – personal, and Vehicle offences. Darkness significantly increases the risk of these crimes occurring, compared with daylight. All other crime categories had odds ratios that did not significantly deviate from one.

Eleven of the fourteen crime categories have 95% CIs that do not overlap with that of at least one other crime category. The crime categories that do not overlap with their CIs are shown in Table 7. Fig 2 shows the odds ratios and 95% CIs for each crime category, also illustrating the fact that the odds ratio CIs for many crime categories do not overlap. This suggests the risk of crime occurring after dark compared with in daylight does differ depending on the category of crime. Note

**Table 7. Counts of crimes in CaseDark, CaseDay, ControlDark and Control Day periods for each crime category and all crime, along with associated odds ratios, 95% CIs and corrected p-values – for main analysis.**

| Crime ID | Crime category | Case-Dark | Case-Day | Contr-olDark | Contr-olDay | Odds ratio | 95% CI | P-value (Holm-Bonferroni corrected) | Other crime IDs the CI interval does not overlap with |
|---|---|---|---|---|---|---|---|---|---|
| 1 | All other theft offences | 979 | 1079 | 1367 | 1674 | 1.11 | 0.99-1.24 | 0.535 | 3,4,5,9,12 |
| 2 | Arson | 111 | 96 | 39 | 54 | 1.60 | 0.98-2.62 | 0.562 | – |
| 3 | Bicycle theft | 178 | 163 | 172 | 265 | 1.68* | 1.26-2.24 | 0.004 | 1,11,13,14 |
| 4 | Burglary | 1268 | 589 | 682 | 817 | 2.58** | 2.24-2.97 | <0.001 | 1,5,6,10,11,13,14 |
| 5 | Criminal damage | 2015 | 1604 | 896 | 1105 | 1.55* | 1.39-1.73 | <0.001 | 1,4,10,11,13,14, |
| 6 | Other sexual offences | 83 | 92 | 76 | 92 | 1.09 | 0.71-1.67 | 1.000 | 4,12 |
| 7 | Rape | 21 | 29 | 11 | 21 | 1.38 | 0.55-3.47 | 1.000 | – |
| 8 | Robbery – business | 57 | 25 | 20 | 17 | 1.94 | 0.87-4.31 | 0.731 | – |
| 9 | Robbery – personal | 213 | 142 | 98 | 122 | 1.87** | 1.33-2.62 | 0.004 | 1,10,11,13,14 |
| 10 | Shoplifting | 973 | 900 | 1638 | 1759 | 1.16 | 1.04-1.30 | 0.086 | 4,5,9,12,14 |
| 11 | Theft from the person | 114 | 114 | 366 | 304 | 0.83 | 0.61-1.12 | 0.996 | 3,4,5,9,12 |
| 12 | Vehicle offences | 929 | 552 | 630 | 750 | 2.00** | 1.73-2.33 | <0.001 | 1,6,10,11,13,14 |
| 13 | Violence with injury | 1313 | 1800 | 914 | 1213 | 0.97 | 0.87-1.08 | 1.000 | 3,4,5,9,12 |
| 14 | Violence without injury | 961 | 1173 | 920 | 1017 | 0.91 | 0.80-1.02 | 0.694 | 3,4,5,9,10,12 |
| **All crime** | | **9219** | **8360** | **7829** | **9210** | **1.30*** | **1.24-1.35** | **<0.001** | |

Notes: * indicates a small effect size, ** indicates a medium effect size, according to the effect size thresholds defined by Olivier & Bell [47]. Effect sizes only reported for odds ratios that significantly deviated from 1.0.

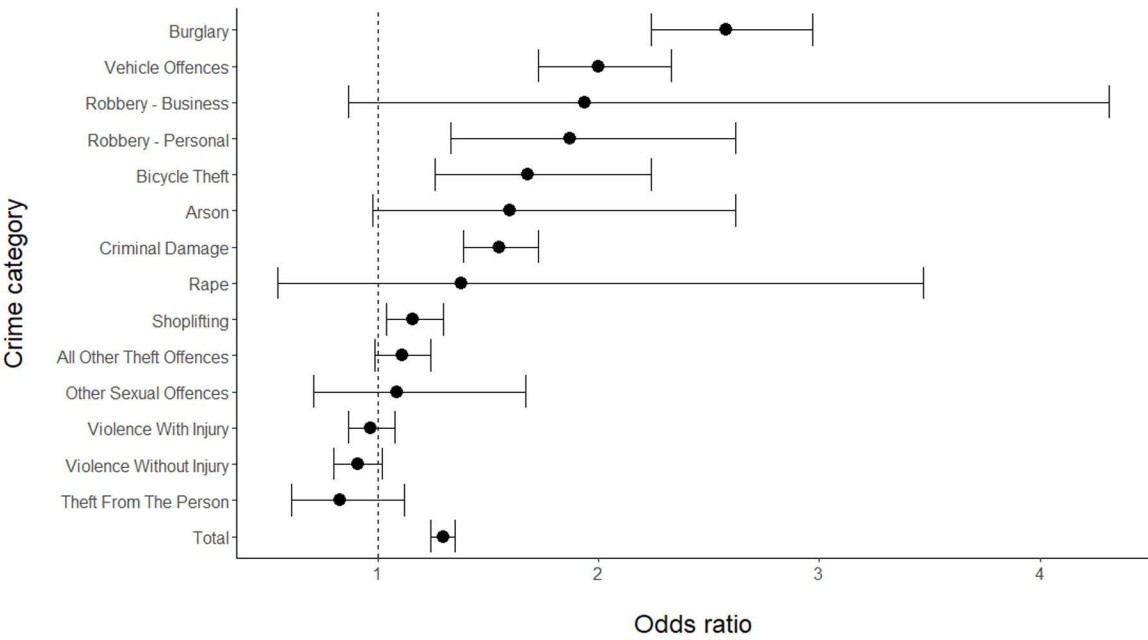

**Fig 2. Odds ratios and 95% CIs for different crime categories, and all crime, for main analysis.**

that some crime categories, e.g., Robbery – business and Rape, have particularly large CIs. This is due to the relatively small counts of these crimes included in their odds ratio calculations.

Odds ratios were also calculated for each MSOA area in South Yorkshire. These were odds ratios for all crime categories combined. Odds ratios were not calculated for each crime category in each MSOA as individual counts for some crime categories may have reached very low numbers, potentially leading to the identification of victims. With such low counts, odds ratios would also have been relatively meaningless due to very large CIs.

There are 172 MSOAs in South Yorkshire. Odds ratios and associated 95% CIs were calculated for each of these. These are shown in Supplementary file S5 File. Twenty five MSOAs had odds ratios with CIs that did not overlap with one. These all had odds ratios above one, and are highlighted in the map of MSOAs in Fig 3. Across all 172 MSOAs, odds ratios ranged between 0.45 (MSOA area E02001609) and 7.00 (MSOA area E02001656).

Due to the relatively small counts of crimes included in the odds ratio calculations for many of the MSOAs, the CIs were often quite large, with a median CI range of 1.88.

One hundred and eight out of the 172 MSOAs had CIs that did not overlap with that of at least one other MSOA, with seven of these MSOAs having CIs that did not overlap with at least ten other MSOAs. This suggests the risk of crime occurring in darkness relative to daylight is different in some MSOAs compared with others.

### 3.2. Sensitivity analysis

We selected three pairs of case and control hours for our main analysis (see Table 2). Other pairings of case and control hours were possible, and our results may be sensitive to the hours selected. We therefore carried out a sensitivity analysis in which we selected different pairings of case and control hours to those used in the main analysis, to test whether the results and conclusions from the main analysis are sensitive to the choice of hours used (see section 2.5).

There were a total of 9,239 crimes recorded in the case and control hours used in the sensitivity analysis, considerably lower than the 34,618 recorded crimes used in the main analysis. This lower volume of crimes led to greater uncertainty

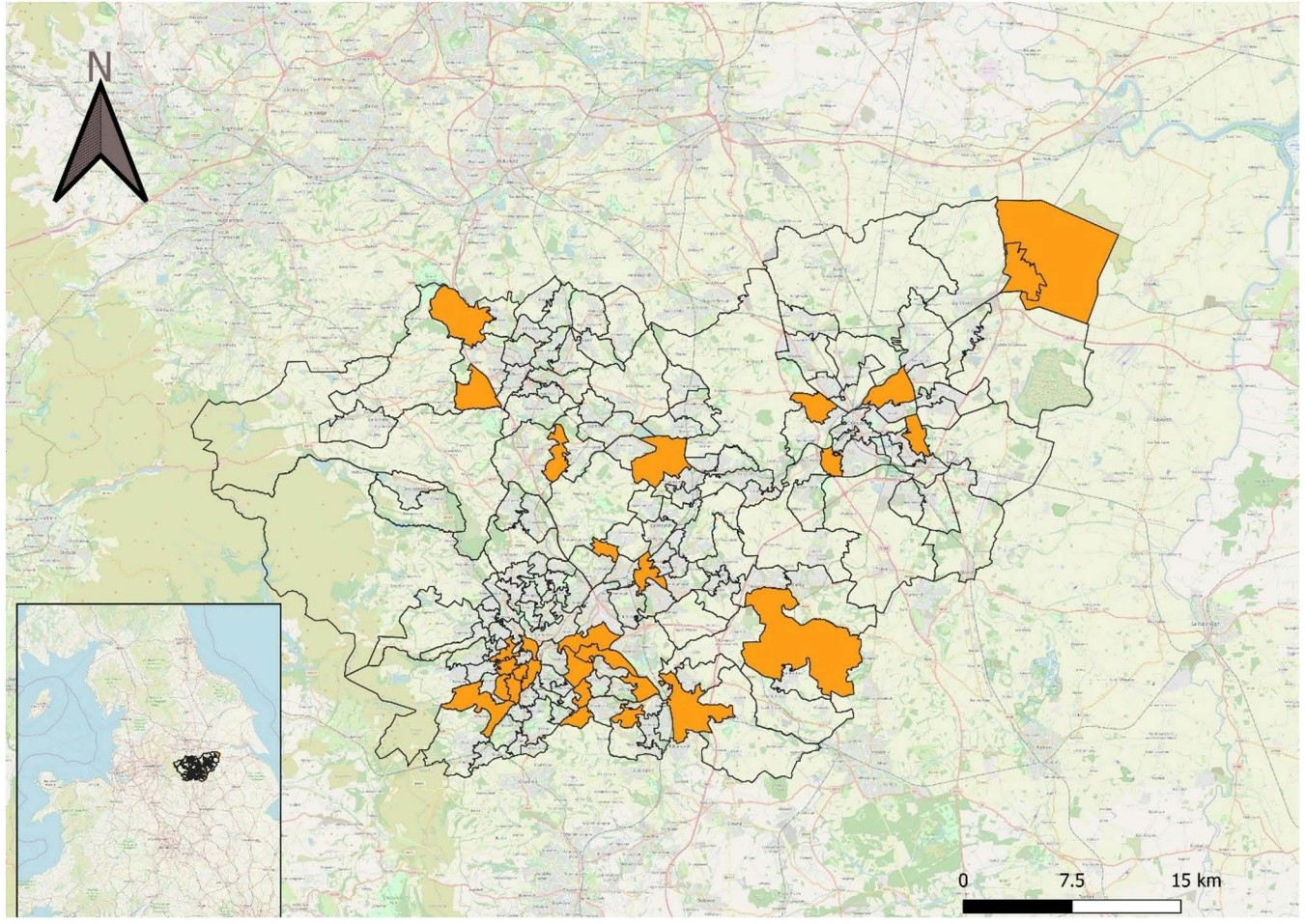

**Fig 3. MSOA boundaries in South Yorkshire.** Those with odds ratio CIs that do not include one are highlighted in orange. All of these MSOAs had odds ratios greater than one. Map imagery reprinted from OpenStreetMap under a CC BY license, with permission under an Open Database License (https://www.openstreetmap.org/copyright/en).

in the odds ratios that were calculated. These odds ratios and their associated 95% CIs and Holm-Bonferroni-corrected p-values are shown in Table 8. Odds ratios and 95% CIs are also shown in Fig 4.

For the sensitivity analysis, the odds ratio for all crime was 1.13 (95% CI: 1.02–1.25). Although the 95% confidence did not overlap with one, the Holm-Bonferroni-corrected p-value for the odds ratio was not significant. The only crime category that showed a significant odds ratio in the sensitivity analysis as well as the main analysis was Burglary. However, two other crime categories had significant odds ratios in the sensitivity analysis but not in the main analysis – Arson (OR: 9.96, 95% CI: 3.10–32.07) and Shoplifting (OR: 0.30, 95% CI: 0.19–0.47).

The sensitivity analysis found that twelve of the fourteen crime categories had 95% CIs that did not overlap with the CI of at least one other crime category. However, there were a number of differences in terms of which crime categories did not overlap with which other categories when comparing the sensitivity analysis with the main analysis. For example, in the main analysis, the CI for Bicycle theft did not overlap with the intervals for four other crime categories. However, in the sensitivity analysis, the CI for Bicycle theft only failed to overlap with one other crime category, and this was not one of four crime categories found in the main analysis.

**Table 8. Counts of crimes in CaseDark, CaseDay, ControlDark and Control Day periods for each crime category and all crime, along with associated odds ratios, 95% CIs and corrected p-values – for sensitivity analysis.**

| Crime ID | Crime category | Case-Dark | Case-Day | Contr-olDark | Contr-olDay | Odds ratio | 95% CI | P-value (Holm-Bonferroni corrected) | Other crime IDs CI interval does not overlap with |
|---|---|---|---|---|---|---|---|---|---|
| 1 | All other theft offences | 158 | 127 | 831 | 641 | 0.96 | 0.74-1.24 | 1.000 | 2,4,10 |
| 2 | Arson | 31 | 7 | 8 | 18 | 9.96*** | 3.10-32.07 | 0.001 | 1,3,4,5,10,12,13,14 |
| 3 | Bicycle theft | 14 | 17 | 71 | 67 | 0.78 | 0.36-1.70 | 1.000 | 2 |
| 4 | Burglary | 233 | 87 | 339 | 273 | 2.16** | 1.61-2.89 | <0.001 | 1,10,12,13,14 |
| 5 | Criminal damage | 197 | 110 | 423 | 310 | 1.31 | 1.00-1.73 | 0.675 | 2,10 |
| 6 | Other sexual offences | 12 | 13 | 18 | 34 | 1.74 | 0.66-4.60 | 1.000 | 10 |
| 7 | Rape | 6 | 5 | 6 | 7 | 1.40 | 0.28-7.02 | 1.000 | – |
| 8 | Robbery – business | 10 | 6 | 9 | 6 | 1.11 | 0.26-4.72 | 1.000 | – |
| 9 | Robbery – personal | 31 | 12 | 34 | 29 | 2.20 | 0.96-5.06 | 0.675 | 10 |
| 10 | Shoplifting | 28 | 70 | 817 | 609 | 0.30*** | 0.19-0.47 | <0.001 | 1,2,4,5,6,9,11,12,13,14 |
| 11 | Theft from the person | 25 | 5 | 272 | 138 | 2.54 | 0.95-6.77 | 0.675 | 10 |
| 12 | Vehicle offences | 148 | 96 | 449 | 277 | 0.95 | 0.71-1.28 | 1.000 | 2,4,10 |
| 13 | Violence with injury | 208 | 130 | 455 | 350 | 1.23 | 0.95-1.60 | 0.917 | 2,4,10 |
| 14 | Violence without injury | 110 | 96 | 457 | 297 | 0.74 | 0.55-1.02 | 0.675 | 2,4,10 |
| **All crime** | | 1212 | 782 | 4189 | 3056 | 1.13 | 1.02-1.25 | 0.219 | |

Notes: * indicates a small effect size, ** indicates a medium effect size, and *** indicates a large effect size, according to the effect size thresholds defined by Olivier & Bell [47]. Effect sizes only reported for odds ratios that significantly deviated from 1.0.

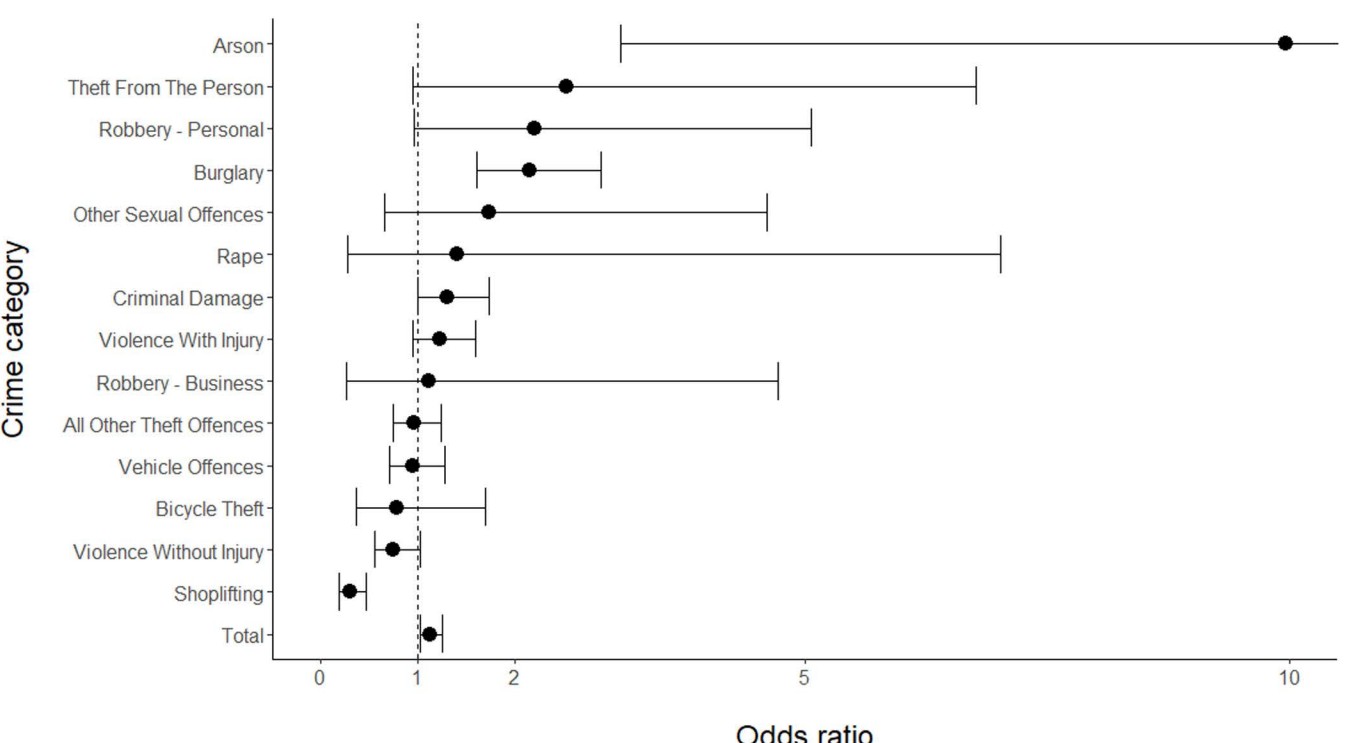

**Fig 4. Odds ratios and 95% CIs for different crime categories, and all crime, for sensitivity analysis.** Note that the upper CI for Arson is truncated.

For the sensitivity analysis, odds ratios were also calculated for each MSOA area in South Yorkshire, as was done in the main analysis. The odds ratios and 95% CIs for each MSOA area are shown in Supplementary file S6–S10 Files. It was not possible to calculate odds ratios for ten of the 172 MSOAs because more than one of the four counts required to calculate the odds ratio was zero (see section 2.4). Of the remaining MSOAs, 42 had 95% CIs that did not overlap with that of at least one other MSOA, and three of these MSOAs had 95% CIs that did not overlap with that of at least ten other MSOAs. There is a smaller number of MSOAs with non-overlapping CIs in the sensitivity analysis than in the main analysis. This is likely due to the larger CIs calculated in the sensitivity as a result of smaller counts of crimes. The median CI range for MSOAs in the main analysis is 1.88, compared with 4.53 in the sensitivity analysis.

Ten of the MSOAs in the sensitivity analysis have 95% CIs that do not overlap with one – six of these have lower CIs that are greater than one and four have upper CIs that are less than one. This compares with 25 MSOAs in the main analysis that had 95% CIs that did not overlap with one, and all of these had lower CIs above one.

## 4. Discussion

Evidence about the relationship between darkness, lighting and perceptions of safety is relatively clear – darkness reduces perceived safety but lighting can offset this and make people feel more reassured to walk after dark [4,48]. Evidence about the relationship between darkness, lighting and actual crime is less clear. Some studies have found no relationship between reduced lighting and crime rates [11] or even a reduction in crime with reduced lighting [49]. A systematic review of literature on this topic found that although street lighting interventions reduced crime rates, these reductions tended to happen in daylight hours as well as after dark [17]. This would suggest reductions in crime were not due to improvements in visibility and lighting conditions, and instead due to an alternative explanation, such as community pride. Such an explanation may be supported by criminological theories such as broken windows theory [50] and social disorganisation theory [51], which suggest signs of crime and disorder, and a lack of informal control through social cohesion, can lead to more crime and disorder. The provision of high quality, well-maintained street lighting could signify an orderly neighbourhood and create a sense of pride in the community, thus deterring criminal activity.

Alternatively, the mechanism for street lighting to influence crime rates could be through changes to visibility levels, rather than as an indication of social orderliness and community pride. This mechanism would align more with an explanation based on routine activity theory and the simultaneous presence of motivated offenders, suitable targets and capable guardians [25]. The lighting in an area may influence the motivation of an offender to commit a crime, e.g., a well-lit area may deter some offenders due to increased risk of exposure and detection. Street lighting could reduce the number of suitable targets available by making them more visible and thus less attractive to offenders, as they may be more likely to be seen. Street lighting could also increase capable guardianship by improving visibility and increasing potential surveillance from bystanders; a well-lit area may also have more people in it [30].

To understand what effect lighting has on crime, and how this effect works, we must first understand how darkness influences the risk of crime occurring. If darkness does not increase the risk of crime it is unlikely that lighting can help reduce crime through mechanisms related to improvements in vision. If we can quantify any increased risk of crime occurring due to darkness, we also then have a metric for assessing the effectiveness of lighting in reducing crime risk after dark.

We therefore carried out an analysis of crime data from the South Yorkshire region of the United Kingdom to assess the influence of darkness on the risk of crime occurring. We used a case/ control method that controlled for the time of day and seasonal influences, similar to the method used in previous work [26–28]. In a development of this work, we introduced a set of decision rules to account for the aoristic nature of many crimes. We also used multiple case and control hours, carried out a sensitivity analysis to assess the impact of our choice of case and control hours, and preregistered the analysis method and our hypotheses prior to accessing the data.

The case/ control method we employed produces odds ratios related to the risk of crime occurring after dark. An odds ratio significantly greater than one indicates a greater risk of crime occurring after dark than during daylight, at the same time of the day.

We tested three hypotheses. Our first hypothesis, that the overall risk of crime in South Yorkshire occurring after dark is greater than during daylight, was supported by our main analysis. The odds ratio for all crimes included in the analysis was 1.30 (95% CI: 1.24–1.35). This was significantly greater than one (p<0.001), suggesting the overall risk of crime occurring is higher after dark than during daylight. The magnitude of this effect of darkness is suggested to be small based on the effect size thresholds for odds ratios proposed by Olivier and Bell [47]. It is, however, larger than the effect of darkness on crime found by Fotios, Robbins and Farrall [27,28]. In their analysis of crime records in three cities in the United States they found an odds ratio for overall crime of 1.05 (95% CI: 1.01–1.10, p=0.03) [27]. In further work extending the number of cities included in their analysis to eleven, the largest odds ratio they found for any city was 1.08 (Chicago, 95% CI: 1.02–1.14, p<0.05). There are a number of potential explanations for the difference in the risk of crime after dark found in the current work and that of Fotios et al. The current work examines crime data in the United Kingdom, whereas Fotios et al examined crime data from the United States. The current work used crime data from across the whole year, whereas Fotios et al examined crime data from the weeks either side of clock changes. There were also differences in the crime categories included in the overall measure of crime risk after dark. However, perhaps the most significant explanation for the difference with previous work is the accounting for aoristic crimes that our current method employs. In their previous work, Fotios et al used the mid-time between the possible start and end times of each crime to determine whether it fell into case or control period. This did not take into account the length of time between the possible start and end times of a crime. In the current analysis, we only included crimes if their time window was one hour or less. This led to a more accurate allocation of crimes into case and control periods, better ensuring a contrast between crimes that occurred in daylight and crimes that occurred in darkness.

Our second hypothesis, that the risk of crime occurring after dark relative to daylight would vary depending on the type of crime, was also supported. Eleven of the fourteen crime categories included in our main analysis had odds ratio 95% CIs that did not overlap with that of at least one other crime category. This suggests the risk of crime occurring after dark does vary depending on the category of crime. We also found that five of the crime categories (Bicycle theft, Burglary, Criminal damage, Robbery – personal, and Vehicle offences) had odds ratios significantly greater than one, suggesting that these crimes are more likely to occur after dark than during daylight, assuming the time of day remains the same. The finding that the risk of personal robbery increases due to darkness agrees with previous research [27,28]. Fotios et al [28] also found that darkness increased the risk of sex offences but we found no effect of darkness on the crime categories of Rape or Other sexual offences. Even when these two categories were combined, to better match the 'Sex offences' category used in Fotios et al [28], an effect of darkness was still not apparent (odds ratio=1.12, 95% CI=0.76–1.64, p=0.57). Our study is unable to explain why five particular crime types (Bicycle theft, Burglary, Criminal damage, Robbery – personal, and Vehicle offences) showed higher risk of being committed after dark than during daylight. One obvious potential explanation though is that perpetrators of these crimes feel less at risk of being detected when visibility conditions are reduced, as occurs after dark. This may not be a factor for other types of crimes, particularly for example if they tend to occur indoors, when outdoor ambient light conditions are less relevant. However, it is also worth noting that the reduction in visibility associated with darkness could have an opposite influence on propensity to offend for some crime types, if better visibility facilitates the crime. For example, thefts from vehicles have been shown to reduce when street lighting is switched off, the assumption being that offenders need a certain amount of light to assess the target of their crime and undertake the crime [49].Our third hypothesis, that the risk of crime occurring after dark varies spatially across the South Yorkshire region, was supported. There was a large range in odds ratios between MSOAs, and 63% of MSOAs had odds ratio CIs that did not overlap with that of at least one other MSOA. Some MSOAs were identified as having a particularly high risk of crime being committed after dark with 25 areas having odds ratio CIs that were above one. To our knowledge

this is the first time that small-scale spatial variations in the risk of crime after dark has been assessed. However, Fotios et al [28] did compare the risk of crime after dark between eleven cities in the United States. There results did not suggest any variation in the risk of crime after dark between these different cities, with the calculated odds ratio CIs all overlapping with each other (see Table 1 in their paper). Our study is unable to provide evidence to explain why areas may vary in the risk of crime occurring after dark. However, a number of possible explanations exist that could be explored in future research. One explanation relates to the presence and type of street lighting in an area. Previous research has suggested a link between street lighting and risk of crime after dark in an area (e.g., [18]). Areas that have less street lighting, or 'poorer quality' lighting (however that might be defined, whether it be in terms of brightness, uniformity or some other characteristic) may be at higher risk of crime being committed after dark. Areas that have more street lighting could cause displacement of after dark crime into neighbouring areas [18,49]. Another possible explanation for spatial variation in the risk of crime after dark is spatial variation in where particular types of crimes occur. As our results show, some crimes are at more risk of being committed after dark than others. If such higher-risk crimes are concentrated in certain areas, this will increase the overall risk of crime being committed after dark in those areas, relative to other areas.

Our work has a number of limitations. Although we attempted to account for the aoristic nature of some crimes by including only those that had a time window of less than one hour in our analysis, it is still possible that we incorrectly categorised crimes into a case or control hour. This may have led to us categorising some crimes as occurring in darkness or daylight when in fact they may have occurred in twilight conditions. For example, a crime with a time window of 17:02–17:59 would have been allocated to the case hour of 17:30–18:29. This would be an incorrect allocation if the crime actually occurred before 17:30, for example, and the attributed light condition for the crime (daylight or darkness, depending on the date it occurred) may have been incorrect. We accept such potential errors as we believe they will be relatively rare and have minimal impact on the overall results and findings. It would be possible to reduce such potential errors by using a smaller time window criterion (For example, only including crimes with a time window of 30 minutes or less in the analysis, rather than one hour), but this would result in a smaller sample of crimes being available.

A further limitation that is related to the aoristic nature of some crimes is that aoristic crimes may be more likely to occur at certain times of day, e.g., when victims are not present (burglaries when the occupants are not at home, for example). Our approach of excluding crimes that had a time window of greater than one hour may therefore have disproportionately affected the number of crimes included in case or control hours. However, this in theory would affect equally the caseDay and caseDark periods (or controlDay and controlDark periods, in the control hour) and we therefore assume does not confound the results in any particular direction. Take burglary as an example. Burglaries may be more likely to occur during the daytime, when occupants are out of the house. This could lead to an underrepresentation of burglaries included in the (daytime) control hours as many might be excluded from the analysis for having a committal time window longer than one hour. We assume this underrepresentation would affect the controlDay and controlDark periods equally however, thus meaning the overall result for burglary is not confounded in a particular direction.

Our analytical approach required the selection of case and control hours. This limited our analysis to only including crimes that potentially occurred during those hours, thus reducing the number of crimes we were able to consider. The selection of the case and control hours also represents a researcher degree of freedom, as it was possible to select other case and control hour pairings to the ones we have used. We attempted to assess the impact of our choice of case and control hours by carrying out a sensitivity analysis that used different pairings of case and control hours. In partial agreement with our main analysis, the sensitivity analysis found some evidence in support of our first hypothesis, as the odds ratio for all crime was greater than one and the CI did not overlap with one, although the Holm-Bonferroni-corrected p-value was not significant.

The sensitivity analysis also found some agreement with the main analysis in the assessment of the second hypothesis, with some crime categories showing a higher risk of committal after dark than others. However, the sensitivity and main analysis were not consistent with which crime categories showed higher risk. Some of this inconsistency could

be explained by the sensitivity analysis producing larger CIs than the main analysis due to the smaller sample of crime counts it was able to utilise. However, it also seems likely that the sensitivity analysis revealed some real differences in the impact of darkness on crime risk, compared with the main analysis. For example, the sensitivity analysis suggested two crime categories that had odds ratios significantly different to one, that were not suggested by the main analysis – Arson and Shoplifting.

The sensitivity analysis found support for our third hypothesis, in agreement with the main analysis. Twenty-six percent of MSOAs had odds ratio CIs that did not overlap with that of at least one other MSOA. Although this is lower than the percentage of non-overlapping MSOAs found in the main analysis (63%), this may be due to the larger CIs obtained in the sensitivity analysis due to the smaller sample of crime counts available. Table 9 compares the evidence found for our three hypotheses from the main and sensitivity analyses.

In agreement with results from the main analysis, results from the sensitivity analysis provided support for all three of our hypotheses. In this respect our findings are robust to the choice of case and control hours. However, there were differences in the detail of the results of the main and sensitivity analysis – for example, the two analyses found differences in the crime categories that were significantly influenced by darkness.

Some of the differences between the main and sensitivity analyses may be due to the smaller sample of crimes available in the sensitivity analysis compared with the main analysis, resulting in larger odds ratio CIs and less confidence in the estimates of the effect of darkness on crime risk. However, it is also possible that the time of day is a factor in the effect darkness has on the risk of a crime occurring. In effect, there may be an interaction between time of day and the influence of darkness. For example, darkness may increase the risk of arson in the morning hours (a statistically significant odds ratio of 9.96 was found for the morning case hours used in the sensitivity analysis) but not in the evening hours (a non-significant odds ratio of 1.60 was found for the evening hours used in the sensitivity analysis).

To ensure any potential confound or interaction effect caused by time of day is accounted for, future research employing the case/ control method used by us and previous studies [27,28] should incorporate all possible case and control hour combinations. A similar approach has been advocated for when applying the case/ control method to travel counts, to assess the impact of darkness on cycling rates [52].

An alternative explanation for the differences found between the main and sensitivity analyses is that the method we have employed is not robust in identifying a consistent effect of darkness and variations found in the odds ratio are essentially just due to noise in the data. One approach to testing whether this is the case is to carry out a 'null' test to check whether the method we have employed correctly returns no effect when there is no change in ambient light condition. This can be done by using case hours that remain in the same ambient light condition throughout the whole year. We would expect there to be no effect (i.e., an odds ratio that is not significantly different to 1.0). We carried out such a null test as a post hoc analysis. The same methods were used as with the main and sensitivity analyses, but case hours were selected that remained in daylight throughout the entire year. Two separate null tests were carried out, with three pairs of

**Table 9. Comparison of evidence in support of our three hypotheses, found in main and sensitivity analyses.**

| Hypothesis | Evidence from main analysis | Evidence from sensitivity analysis |
|---|---|---|
| 1. Overall risk of crime occurring after dark is greater than during daylight | Supported – OR for all crime is significantly greater than 1.0 | Part Supported – OR for all crime is not significantly greater than 1.0, but lower 95% CI is above 1.0 |
| 2. Risk of crime occurring after dark relative to daylight varies depending on type of crime | Supported – 11 of the fourteen crime categories have OR 95% CIs that do not overlap with those of at least one other crime category | Supported – 12 of the fourteen crime categories have OR 95% CIs that do not overlap with those of at least one other crime category |
| 3. Risk of crime occurring after dark relative to daylight is not uniform across MSOAs | Supported – 108 MSOAs had OR 95% CIs that did not overlap with those of at least one other MSOA | Supported – 42 MSOAs had OR 95% CIs that did not overlap with those of at least one other MSOA |

case and control hours used in each (see Table 10). The same periods of the year used in the main analysis to define the daylight and dark periods were also used in both null tests. Fig 5 shows the odds ratios and associated confidence intervals for each crime category for null tests 1 and 2. All confidence intervals were very close to or overlapping with 1.0, and holm-bonferroni p-values confirmed that no crime categories had an odds ratio significantly different to 1.0. This confirms that no effect is found when case hours with a constant ambient light condition are used, providing supporting evidence to the assumption that our case and control method is appropriately detecting effects of darkness on crime, where present.

**Table 10. Case and Control hour pairs used in null test 1 and 2, and days of year used to define periods in the odds ratio equation.**

| Null test 1 | | | |
| --- | --- | --- | --- |
| Case hour | Control hour | Days of year used to define 'CaseDark' and 'ControlDark' periods | Days of year used to define 'ControlDay' and 'ControlDark' periods |
| 10:00-10:59 | 13:00-13:59 | 1-33, 304–365 (or 366 if leap year) | 87-278 |
| 11:00-11:59 | 14:00-14:59 | 1-65, 296–365 (or 366 if leap year) | 90-253 |
| 12:00-12:59 | 15:00-15:59 | 1-83, 270–365 (or 366 if leap year) | 120-227 |
| Null test 2 | | | |
| Case hour | Control hour | Days of year used to define 'CaseDark' and 'ControlDark' periods | Days of year used to define 'ControlDay' and 'ControlDark' periods |
| 13:00-13:59 | 10:00-10:59 | 1-33, 304–365 (or 366 if leap year) | 87-278 |
| 14:00-14:59 | 11:00-11:59 | 1-65, 296–365 (or 366 if leap year) | 90-253 |
| 15:00-15:59 | 12:00-12:59 | 1-83, 270–365 (or 366 if leap year) | 120-227 |

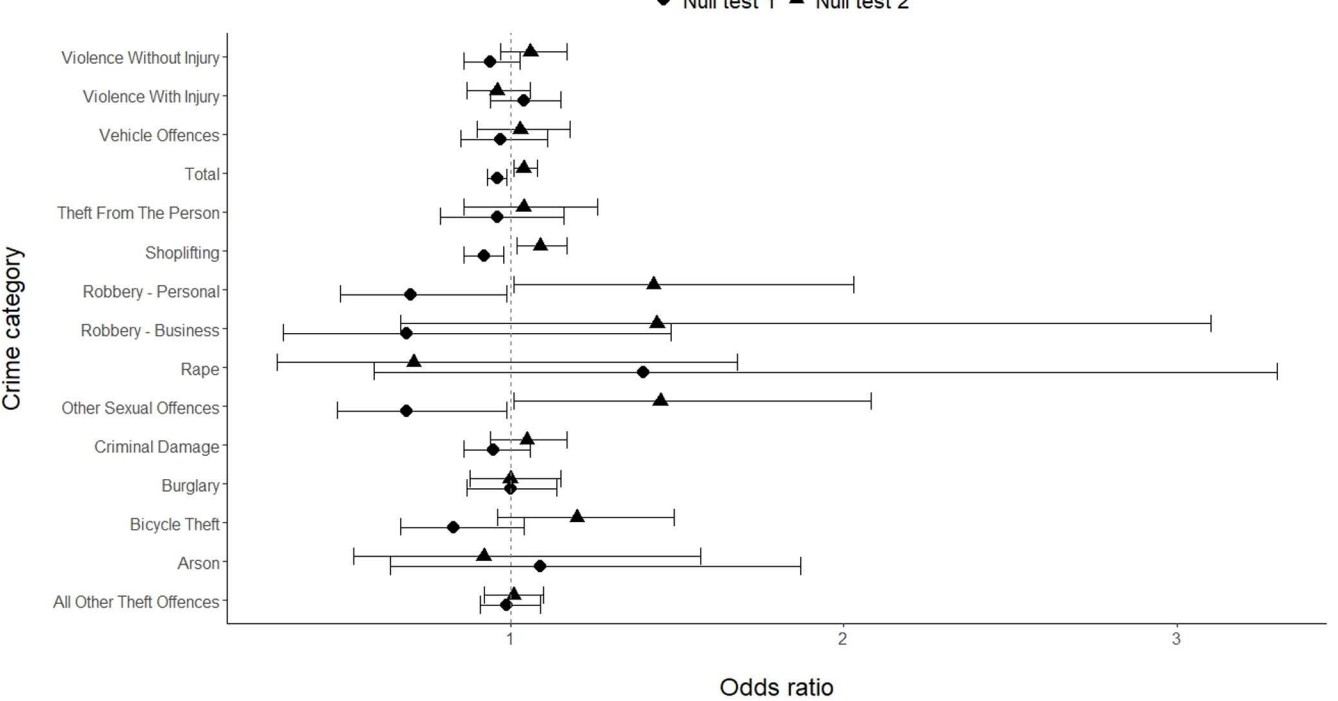

**Fig 5. Odds ratios and 95% CIs for different crime categories, and all crime, for null tests.**

## 5. Conclusion

We employed a case and control hour approach to assess the impact of darkness on crime whilst controlling for the influence of time of day. Our study built on previous work using this approach [27,28] by using data from the UK rather than the United States, using a whole-year rather than clock-change approach to increase sample sizes, and more accurately accounting for aoristic crimes. We also developed our hypotheses and analytical plan without any preliminary sight of the data, preregistering the study to increase transparency and strengthen the validity of our conclusions [53].

We tested three hypotheses, and found evidence in support of all three. First, we found that the overall risk of crime occurring after dark was higher than in daylight, after controlling for the time of day. Second, we found that the risk of crime occurring after dark compared with during daylight varied depending on the crime category. Our main analysis found that Bicycle theft, Burglary, Criminal damage, Robbery – personal, and Vehicle offences were crime categories at greater risk of being committed after dark than during daylight. Third, we found that the overall risk of crime occurring after dark compared with daylight varied spatially across the South Yorkshire region, with some MSOAs having higher risk of after dark crime than others.

These findings provide strong evidence that darkness influences the risk of crime. This suggests that street lighting could potentially have a mediating role on the risk of crime but this requires further confirmation. Future work should investigate how the presence of lighting, and the characteristics (e.g., illuminance and uniformity) of that lighting, influence the risk of crime. Such evidence can be used to optimise lighting, after balancing against negative impacts of lighting such as energy and carbon costs, light pollution, and harmful effects on flora and fauna. Any future work should also account for any potential displacement effects, as previous work has suggested light conditions in one area can influence the committal of crime in adjacent areas, through the spatial displacement of criminal activity [18,49]. The method used in our current study could be harnessed to provide a metric for the effectiveness of lighting in reducing the risk of crime after dark, and to assess displacement effects between adjacent areas.

## Supporting information

**S1 File. R code for analysis of crime data.**
(R)

**S2 File. Synthetic crime data (from CMS system).**
(CSV)

**S3 File. Synthetic crime data (from Connect system).**
(CSV)

**S4 File. Output data by crime category – main analysis.**
(CSV)

**S5 File. Output data by crime category – sensitivity analysis.**
(CSV)

**S6 File. Output data by MSOA – main analysis.**
(CSV)

**S7 File. Output data by MSOA – sensitivity analysis.**
(CSV)

**S8 File. R code for null test analyses.**
(R)

**S9 File. Output data for null test 1.**
(CSV)

**S10 File. Output data for null test 2.**
(CSV)

## Acknowledgments

The authors wish to acknowledge the support and contribution made to this work by Ray Froggatt, South Yorkshire Police.

## Author contributions

**Conceptualization:** Jim Uttley.

**Data curation:** Rosie Canwell, Jamie Smith, Sarah Falconer.

**Formal analysis:** Jim Uttley, Rosie Canwell, Sarah Falconer.

**Methodology:** Jim Uttley, Steve Fotios.

**Project administration:** Jim Uttley, Jamie Smith.

**Resources:** Jim Uttley, Jamie Smith.

**Supervision:** Jamie Smith, Steve Fotios.

**Visualization:** Jim Uttley.

**Writing – original draft:** Jim Uttley, Steve Fotios.

**Writing – review & editing:** Jim Uttley, Rosie Canwell, Jamie Smith, Sarah Falconer, Yichong Mao, Steve Fotios.

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
