## [Decision Letter · Decision Letter 0]

4 Jul 2024

PONE-D-24-09508Does darkness increase the risk of certain types of crime? A Registered Report ArticlePLOS ONE

Dear Dr. Uttley,

Thank you for submitting your manuscript to PLOS ONE. After careful consideration, we feel that it has merit but does not fully meet PLOS ONE’s publication criteria as it currently stands. Therefore, we invite you to submit a revised version of the manuscript that addresses the points raised during the review process.

We look forward to receiving your revised manuscript.

Kind regards,

Jesús Espinal-Enríquez

Academic Editor

PLOS ONE

Journal Requirements:

https://journals.plos.org/plosone/s/file?id=ba62/PLOSOne_formatting_sample_title_authors_affiliations.pdf"

2. We note that Figure 3 in your submission contain [map/satellite] images which may be copyrighted. All PLOS content is published under the Creative Commons Attribution License (CC BY 4.0), which means that the manuscript, images, and Supporting Information files will be freely available online, and any third party is permitted to access, download, copy, distribute, and use these materials in any way, even commercially, with proper attribution. For these reasons, we cannot publish previously copyrighted maps or satellite images created using proprietary data, such as Google software (Google Maps, Street View, and Earth). For more information, see our copyright guidelines: http://journals.plos.org/plosone/s/licenses-and-copyright.

1. You may seek permission from the original copyright holder of Figure 3 to publish the content specifically under the CC BY 4.0 license.  

Additional Editor Comments

Dear Dr. Uttley,

your manuscript has been carefully assessed by four different reviewers. I decided to ask you for a major revision since there are several minor comments from all reviewers. Please take into account all comments to elaborate a revised version of your manuscript. Despite all reviewers find your work interesting and useful, there are some concerns that should be addressed before acceptance.

Reviewers' comments:

Reviewer's Responses to Questions

**Comments to the Author**

1. Does the manuscript adhere to the experimental procedures and analyses described in the Registered Report Protocol?

If the manuscript reports any deviations from the planned experimental procedures and analyses, those must be reasonable and adequately justified.

Reviewer #1: Yes

Reviewer #2: Yes

Reviewer #3: Yes

Reviewer #4: Yes

2. If the manuscript reports exploratory analyses or experimental procedures not outlined in the original Registered Report Protocol, are these reasonable, justified and methodologically sound?

A Registered Report may include valid exploratory analyses not previously outlined in the Registered Report Protocol, as long as they are described as such.

Reviewer #1: Yes

Reviewer #2: Yes

Reviewer #3: Yes

Reviewer #4: Yes

3. Are the conclusions supported by the data and do they address the research question presented in the Registered Report Protocol?

The manuscript must describe a technically sound piece of scientific research with data that supports the conclusions. The conclusions must be drawn appropriately based on the research question(s) outlined in the Registered Report Protocol and on the data presented.

Reviewer #1: Yes

Reviewer #2: Yes

Reviewer #3: Partly

Reviewer #4: Yes

4. Have the authors made all data underlying the findings in their manuscript fully available?

Reviewer #1: Yes

Reviewer #2: Yes

Reviewer #3: No

Reviewer #4: No

5. Is the manuscript presented in an intelligible fashion and written in standard English?

Reviewer #1: Yes

Reviewer #2: Yes

Reviewer #3: No

Reviewer #4: Yes

6. Review Comments to the Author

Please use the space provided to explain your answers to the questions above. (Please upload your review as an attachment if it exceeds 20,000 characters)

Reviewer #1: In response to the shortcomings of existing research, this article conducts a more detailed study on whether darkness increases the risk of certain types of crime. This article divides the research time period into Case hour and Control hour, and Case hour is divided into CaseDark and CaseDay while Control hour is divided into ControlDark and ControlDay. Then, the article studies the impact of different time periods on different types of crimes, and conducts main analysis and sensitivity analysis; The authors also studied the different impacts of darkness on crime situations in different regions. Overall, the research in this article is comprehensive and detailed, and the hypothesis of this article has been proven through experiments and analysis.

However, there are still some problems with the article that require further revision and improvement by the authors. Specific suggestions are as follows:

1、 What are the three hypotheses of Article based on? Please provide an explanation.

2、Is there a reference basis for the calculation formula of OddsRatio and 95% CI? If so, please list them out; If not, please explain the reason for this calculation.

3、The experimental section did not mention the model selection for regression analysis. If a model is used for regression analysis, please introduce the model and its selection criteria; If not, please provide a detailed introduction to the analysis methods in the experimental section.

4、There are differences between the results of the main analysis and sensitivity analysis. For areas of inconsistency, should the conclusion be based on the main analysis or sensitivity analysis?

5、The paper states that the results indicate that lighting may help reduce the risk of crime after dark, but the article studies the impact of day and night on crime and does not experimentally prove the impact of lighting on crime (natural light and artificial lighting are different).

Reviewer #2: Study Overview: With crime data from South Yorkshire (UK), this study used a case-control method to test the impact of ambient light on crime counts. Leveraging shifts in daylight saving times, crime counts were compared for times in daylight and darkness at different times of the year (case) vs. times always in daylight (control). Analyses were further broken out by offense type and South Yorkshire regions. The analysis most closely resembles that of Fotios, Robbins, and Farrall (2021). Findings show statistically significant different odds ratios overall, for certain offense types, and certain regions.

General thoughts: The study addresses an interesting research question, the link between ambient light and crime, and has practical implications for practitioners, e.g., allocation of law enforcement resources, ecological interventions involving lighting. The design exploits sharp discontinuities in light/dark due to daylight saving, a plausible identification strategy. The methodological approach is sound: the research questions were pre-registered, past methodological approaches were followed, and various sensitivity checks were carried out. Importantly, the study makes a substantial contribution to the literature by extending Fotios et al. to a different region and more carefully dealing with aoristic crimes (ie, crimes that occurred at an indefinite time). The paper is organized well and clearly written.

Major suggestions:

1. The authors make several references to competing ideas for why ambient light might be associated with crime (eg, community pride, routine changes). Early on in the introduction, it would be useful if these ideas were formally laid out. The idea of community pride falls neatly into a couple well-established criminological theories: broken windows theory (first spelled out by Wilson and Kelling) and social disorganization theory (see Shaw and McKay; specifically, factors related to the physical condition of a neighborhood). Findings showing that improved lighting reduces crime might be better explained by signs of improved order in a neighborhood rather than light. Notably, this study can be interpreted as evidence in favor of ambient light as the key ingredient in those interventions, as the variable of light was carefully isolated here. Likewise, routine activity theory might also provide an alternative causal mechanism for the impact of light on crime. This theory is alluded to in several places, but again formally describing it early on will be helpful, as well as specifically discussing how the various elements (motivated offenders, suitable targets, capable guardians) might be implicated.

2. One big gap in the paper is the lack of an explanation for why ambient light had an effect only on certain offense types (in the main analysis: bike theft, burglary, criminal damage, robbery, vehicle offenses). While I appreciate the authors hesitancy to speculate, in this case I believe it would strengthen the paper to suggest reasons as to why these patterns emerged. Similarly, beyond showing that only certain regions were impacted by changes in light, there does not appear to be an explanation for this finding either. Were there any observable differences in the regions where an effect was found that might explain the finding?

3. Suggested sensitivity checks: I was not fully convinced that the approach of using the entire year of data was superior to Fotios et al’s approach of only using data from around the change in daylight saving times. Given the difference in effect sizes between the two papers, I’d be interested to see how much of this was explained by the difference in study timeframes. One possible way to get at this would be to also run your analysis restricted to the periods used in Fotios. Another concern I had was the substantial difference you found when looking at afternoon hours vs morning hours (sensitivity test with different case and control hours). In particular, it made me worry that the distribution might be sufficiently variable throughout the day that your analysis could simple be picking up noise. To address this, it would be useful to conduct a placebo test, ie, run the analysis with ‘case’ hours in daylight throughout the year. A null effect here would be reassuring.

Minor suggestions:

1. The reader would benefit from additional subheadings in the introduction section, eg, limitations of past research on p. 6; current focus on p. 10

2. One of the major findings from research on the impact of lighting on crime is displacement (eg, see Chalfin, Kaplan, Laforest, 2020). In the discussion section, it would be useful to revisit the issue of displacement in light of your findings (pun intended). Does this study tell us anything useful about the strength of those findings? About how to deal with displacement?

3. It’s clear that the authors have specialized knowledge about light. For other researchers/policymakers, it might be useful for the authors to spell out with technical detail the minimal dose of light that would be expected to be effective, given the findings here from a natural setting.

Reviewer #3: This peer review is of the manuscript entitled “Does darkness increase the risk of certain types of crime? A Registered Report Article”. The study uses police-recorded crime data from a UK police agency to test three hypotheses relating to the relationship between darkness and various crime types.

I agreed to review this article because the topic has fascinated me for years. I admire the methods used and believe this work to have used sound analytical decisions and be able to contribute to the field of knowledge. I particularly like that the unit of analysis is space and different crime categories, which is a strength of this research.

However, I believe this manuscript needs to be significantly revised to enable readers to grasp the true value of the findings. In what follows I outline my substantive comments, before passing over to my minor comments. I would like to see this work published so my comments are in the spirit of striving to be constructive.

Substantive comments

My gravest concern in this manuscript is the lack of adherence to scientific conventions. For example, the authors talk about ‘confirming’ their hypotheses. The philosophy of science is clear (see classic work by Popper) that hypotheses cannot be confirmed. They can only be supported, partially so in many cases. Indeed, the job of the scientist is to try to falsify the hypothesis; although many scientists are not particularly scientific in their conduct. My recommendation is that the authors acquaint themselves with the philosophy of Popperian scientific principles and attenuate the language they use accordingly with regards to hypotheses.

Relatedly, the purpose of sensitivity testing is to judge whether the findings are predominantly the result of analytical decisions made in the research process. Since the research reported in this research does, indeed, find that results change according to what hour is chosen for the control group, this suggests either that a) patterns are artefacts of analytical decisions rather than of true relationships or b) the control hours used in the sensitivity analysis introduce confounding variables into the analysis. Only when sensitivity analyses do not appreciably affect the findings can you be confident that the original findings are not flukes.

Another hallmark of science is that science writing should be precise and concise. This is a developmental skill that is honed over one’s career. I’m reminded of the pithy observation: “If I had more time I would have written you a shorter letter”. The aspiration in scientific writing is to ‘say it right, say it once’, not to repeat. The manuscript I reviewed was severely overwritten, with needless repetition throughout and tangential information (e.g., illuminance under twilight conditions is just one example) included. Omitting twilight conditions in the analysis was probably wise, so to have an extended discussion about twilight is puzzling. Likewise, the discussion about crime recoding processes can be condensed into one sentence. The same applies to data excluded from analysis. In Section 2.7 all you have to say here is that a coding script was written a priori based on knowledge of the data structure and passed to the data owners to run. There are countless other examples where important points can be condensed into one sentence and severely condense the manuscript.

If this work started life as a (laudible) student project then time now should be invested in refining the writing to remove redundancy and to make every word in every sentence be doing the heavy lifting. I believe the current manuscript could be reduced by two-thirds in words and still communicate the same crucial information. You may like to consult a writing book (e.g., my current favourite is John Eck’s ‘Writing with sweet clarity) to help you to edit your work until it is clear and concise.

To elaborate, many paragraphs in the literature review, and elsewhere, can be reduced to a sentence or two. For example, the collective findings of studies that examine darkness as a feature of fear of crime. It would be entirely acceptable here to just say “In summary, a range of studies using different methods have demonstrated perceptions of safety tend to be lower (and fear of crime higher) after dark than during daylight, and when an area is less well lit after dark.” And provide several citations and thus delete the extraneous information.

A further hallmark of science is that inferences are used in reasoning. I am confused as to why ‘fear of crime’ is discussed as thoroughly as it is in the manuscript. Fear of crime is important and the vastness of the literature is not adequately covered, but the relation of fear of crime to crime (the topic of inquiry) is nebulous. For example, the premises of why darkness is likely to inspire fear of crime are not logically linked to the conclusion that darkness does indeed inspire fear of crime. And this is not linked to crime occurrence in any way. Either specify the causal relationship between fear of crime and crime more clearly or de-emphasise the fear of crime narrative. In my view fear of crime is tangential and a scientific paper only focuses on that which is central.

Scientific writing also requires all claims to be substantiated. Examples of where this has not been done are:

• Page 12: street lighting consumes energy and impacts on nocturnal environment.

• Page 12: it is not clear how land use, economic conditions and temporal factors have been chosen to illustrate the point of confounders to lighting influences. You need to articulate the argument more logically for the reader to be able to follow.

Minor points for consideration

Synthetic data is the conventional term for what you call ‘fictional data’. There are contemporary criminologists writing about this whose work would be worth reading so you understand the implications of providing such data.

Page 10, line 102, reference [13] used precise crime data and hence undermines the trend you present earlier in the manuscript.

Given the mention of economic and environmental concerns of lighting earlier in the manuscript, I was surprised that the following sentence did not acknowledge these trade-offs on page 15: “Good lighting could be considered that which offsets any increase in criminal activity after dark.”

Page 16, lines 297-302, these justifications should probably be in the methods section.

Table 1: Would potentially make this clearer at first glance if the case and control hours/times were in the table.

Table 3: I do not think this table necessary. You could just say crime data included the offence code, the MSOA code/name, the committed from/to date and time—and how any variables were transformed—instead. In one sentence.

Page 24: It is not clear here how Fisher's exact test is being applied. is it for the 2x2 table of case/control light/dark? A little more foregrounding of this test would be useful in the methods section.

Page 18, 31: it is not clear why one would not want confidence intervals to overlap. This needs a logical argument, expressed using statistical theory, to explain why it matters.

It would be useful context to know what proportion of crime fell into each of the case and control hours used in the analysis prior to Table 1.

Page 26: I did not understand the sentence: The proportion of crimes in the control hour when it was in darkness was 50% of all crimes recorded in the control hour. Maybe rephrase?

Cite the protocol document when you first mention it in the manuscript. In addition, consider referring to the protocol document throughout for information that is contained there, so your registered report just documents the new information. Presently, there is a lot of self-plagiarism between the protocol and registered report.

Figure 2: a vertical line at the 1 marker would help intuitive interpretation of this plot.

Page 31: you might benefit from leading with the finding: 25 MSOAs had odds ratios with CIs that did not overlap with one. These all had odds ratios above one and are highlighted in the map of MSOAs in Figure 3.

Page 32: As a reader I am left wondering what is it about these MSOAs that mean darkness facilitates crime? This is a novel part of the analysis and deserves more attention. What land use/populations/conditions are creating situations there that are encouraging crime? Presently, all you produce is descriptives, not explanations for why darkness might be important at these locations in particular. What do they have in common?

Figure 3: legend, scale bar, north arrow, inset map regarding study area? These are all essential cartographic features of a map.

Page 35: is it worth saying that shoplifting is predominantly committed in the daytime (according to shoplifters the early afternoon hours are the perfect level of footfall in shops for optimal conditions for shoplifting) and this may explain your results?

Page 37: I’m not convinced you can call your treatment of aoristic crime a ‘method’. To my mind it is a set of decision rules for analysis.

Page 40: I do not share your confidence that excluding crimes that were reported over 1 hour will not upwardly or downwardly bias your results. I imagine this will play out differently over crime types. For example, going on international trends it is likely that the daytime period is when people report burglary, but the nighttime when they report vehicle crime has happened. If this holds in your data, these patterns would bias your results but cancel each other out in aggregate. I would like to see a fuller analytic exploration of this before making a sweeping statement that “we therefore assume does not confound the results in any particular direction.”

Reviewer #4: The authors present a very comprehensive examination of how daylight influences crime. There is one limitation/framing issue that I would like to see the authors consider ahead of publication. The lighting of the area that the specific crimes took place in was not measured, only the level of daylight was accounted for. This is important as much of the prior literature cited, and the implications discussed, consider street lighting. The current study is therefore not so interested in the risk factor of darkness as it is the protective factor of daylight. This distinction between daylight, light, and darkness should be explored more in the discussion in particular.

7. PLOS authors have the option to publish the peer review history of their article (what does this mean? ). If published, this will include your full peer review and any attached files.

**Do you want your identity to be public for this peer review?** For information about this choice, including consent withdrawal, please see our Privacy Policy .

Reviewer #1: No

Reviewer #2: No

Reviewer #3: No

Reviewer #4: No

---

## [Author Response · Author response to Decision Letter 1]

4 Mar 2025

Please see submitted document "Response to reviewer comments - SUBMITTED", for our response to all reviewer comments.

---

## [Decision Letter · Decision Letter 1]

22 Apr 2025

Does darkness increase the risk of certain types of crime? A Registered Report Article

PONE-D-24-09508R1

Dear Dr. Uttley,

We’re pleased to inform you that your manuscript has been judged scientifically suitable for publication and will be formally accepted for publication once it meets all outstanding technical requirements.

Kind regards,

Jesús Espinal-Enríquez

Academic Editor

PLOS ONE

Additional Editor Comments (optional):

Dear Dr. Uttley,

I am pleased to inform you that your manuscript titled "Does darkness increase the risk of certain types of crime? A Registered Report Article" has been accepted for publication. Your work represents a significant contribution to the debate on the relationship between lighting and crime, standing out for its methodological rigor, access to individual crime records, and comprehensive analytical approach.

During the review process, some reviewers praised the depth of the study and its careful execution. However, one of the reviewers raised concerns about how the relationship between light and crime is conceptualized, noting that the study measures the impact of natural light but does not account for artificial lighting conditions in the areas where crimes occurred. While I find your response to this concern reasoned and coherent with your study objectives, I believe the comment highlights a complementary perspective worth exploring.

Considering the length of the review process and the overall strength of the manuscript, I have decided not to request further revisions at this stage. However, I encourage you to consider this perspective in future work. In particular, the inclusion of measurements or estimates of artificial lighting conditions may help to further nuance the analysis of environmental effects on crime and strengthen future contributions to this area.

Congratulations on this excellent work. I look forward to seeing its impact on the field.

Best regards,

Jesús

Reviewers' comments:

Reviewer's Responses to Questions

**Comments to the Author**

1. Does the manuscript adhere to the experimental procedures and analyses described in the Registered Report Protocol?

If the manuscript reports any deviations from the planned experimental procedures and analyses, those must be reasonable and adequately justified.

Reviewer #2: Yes

Reviewer #3: Yes

Reviewer #4: Yes

2. If the manuscript reports exploratory analyses or experimental procedures not outlined in the original Registered Report Protocol, are these reasonable, justified and methodologically sound?

A Registered Report may include valid exploratory analyses not previously outlined in the Registered Report Protocol, as long as they are described as such.

Reviewer #2: Yes

Reviewer #3: Yes

Reviewer #4: Yes

3. Are the conclusions supported by the data and do they address the research question presented in the Registered Report Protocol?

The manuscript must describe a technically sound piece of scientific research with data that supports the conclusions. The conclusions must be drawn appropriately based on the research question(s) outlined in the Registered Report Protocol and on the data presented.

Reviewer #2: Yes

Reviewer #3: Yes

Reviewer #4: No

4. Have the authors made all data underlying the findings in their manuscript fully available?

Reviewer #2: Yes

Reviewer #3: Yes

Reviewer #4: No

5. Is the manuscript presented in an intelligible fashion and written in standard English?

Reviewer #2: Yes

Reviewer #3: Yes

Reviewer #4: Yes

6. Review Comments to the Author

Please use the space provided to explain your answers to the questions above. (Please upload your review as an attachment if it exceeds 20,000 characters)

Reviewer #2: In my initial review, I was impressed with the overall quality of the paper and recommended acceptance with minor revisions. Nearly all of my suggested revisions have been addressed in the updated manuscript. The paper is now more firmly grounded in criminological theory (e.g., routine activity, broken windows). The authors have conducted additional "placebo" checks, and the null findings further support a causal interpretation. Additionally, several other recommendations were incorporated, such as addressing the possibility of displacement as an alternative explanation.

Where the authors did not implement suggested changes, they provided adequate justification. For instance, they explained their decision not to apply the Fotios et al. approach of restricting data to times around daylight saving, as doing so would have significantly deviated from the pre-registered plan.

Given these improvements, I recommend the paper for publication.

Reviewer #3: The authors have addressed as many of the Reviewer's concerns as they could within the constraints of not being able to change anything in the Introduction and Methods. The additional analyses completed and the null effects found for the 'placebo test' are reassuring.

Reviewer #4: The authors have chosen not to make any changes in regard to my comment that the text is actually interested in the protective factor of daylight rather than the influence of darkness. This is because the authors do not measure environmental lighting conditions, rather they simply factor for the hours of sunlight. The authors have not factored in any artificial lighting (such as street lighting). This is not an issue of framing, this is an issue of what was actually analyzed in the manuscript. The authors may have goals for their research regarding darkness, they have failed to achieve those goals through their current methodology.

7. PLOS authors have the option to publish the peer review history of their article (what does this mean? ). If published, this will include your full peer review and any attached files.

**Do you want your identity to be public for this peer review?** For information about this choice, including consent withdrawal, please see our Privacy Policy .

Reviewer #2: **Yes: ** Stephen Koppel

Reviewer #3: No

Reviewer #4: No

---

## [Editor Report · Acceptance letter]

PONE-D-24-09508R1

PLOS ONE

Dear Dr. Uttley,

I'm pleased to inform you that your manuscript has been deemed suitable for publication in PLOS ONE. Congratulations! Your manuscript is now being handed over to our production team.

Kind regards,

on behalf of

Dr. Jesús Espinal-Enríquez

Academic Editor

PLOS ONE